# Genome-wide identification and analysis of recurring patterns of epigenetic variation across individuals
Jennifer Zou[1,7], Emily Maciejewski[1,2,7] & Jason Ernst [1,2,3,4,5,6] ✉

Epigenetic mapping studies across individuals have identified many positions of epigenetic variation across the human genome. However the relationships between these positions, and in particular global patterns that recur in many regions of the genome, remains understudied. In this study, we use a stacked chromatin state model to systematically learn global patterns of epigenetic variation across individuals and annotate the human genome based on them. We apply this framework to histone modification data across individuals in lymphoblastoid cell lines and across autism spectrum disorder cases and controls in prefrontal cortex tissue. We find that global patterns are correlated across multiple histone modifications and with gene expression. We use the global patterns as a framework to predict trans-regulators and study a complex disorder. The frameworks for identifying and analyzing global patterns of epigenetic variation are general and we expect will be useful in other systems.

Understanding molecular variation is fundamental to understanding variation in complex traits. Many studies have identified molecular variation across individuals in transcription factor (TF) binding, gene expression, histone modifications, and other molecular phenotypes[1–7].

Understanding variation in histone modifications that are associated with enhancers or promoters can be of particular interest since variants for many diseases and other phenotypes are enriched in enhancer or promoter regions of the genome[8–12]. A number of studies have mapped histone modifications associated with enhancers and promoters across many individuals[3–6,13–16]. These studies have identified thousands of regions where histone modifications differ across individuals.

These previous studies often identify a set of consensus regions across individuals with histone modification signal, such as merged peaks, and perform a marginal association test between each region to an external data set. For example, in histone quantitative trait loci studies, variation is identified by associating histone modification signal across individuals with genetic variants. Similarly, for differential peak analysis, variation in a single region is associated with an external label, such as cases and controls[4,17–19]. An approach that allows for joint analysis of multiple marks is to learn combinatorial and spatial patterns of epigenetic marks (termed chromatin states)[20] using methods such as ChromHMM[21], which are then associated with different candidate biological interpretations. In the presence of data from multiple cell types, a common approach is to apply ChromHMM using the concatenated approach[22] where through a virtual concatenation of multiple cell types a common set of state definitions are learned and used to annotate each cell type. This approach has been previously adapted to histone modification data across multiple individuals from the same cell type by virtually concatenating data across individuals for each data type and learning a chromatin state model using the ChromHMM software[3,21]. Under this approach, individual-specific genome chromatin state annotations were generated that contain chromatin state assignments for each individual genome-wide. These annotations were then used post-hoc to identify regions with variable chromatin states across individuals[3]. While informative, one understudied aspect of these previous approaches is the relationships between the variable regions, in particular recurring patterns of epigenetic variation across individuals observed in many regions of the genome.

The concatenated approach of the ChromHMM framework, while modeling combinatorial patterns across individuals, does not model any recurring patterns across the genome as each individual is given a unique genome-wide chromatin state annotation. A potential alternative to this concatenated approach is applying the stacked modeling approach of ChromHMM[22]. Under this modeling approach, instead of virtually concatenating the data from multiple individuals, the histone data from multiple individuals is stacked and viewed as separate inputs. This results in a single model based on patterns across all individuals and marks and a single genome-wide annotation shared across all individuals. This stacked

---

[1]Computer Science Department, University of California, Los Angeles, Los Angeles, CA, USA. [2]Biological Chemistry Department, University of California, Los Angeles, Los Angeles, CA, USA. [3]Department of Computational Medicine, University of California, Los Angeles, Los Angeles, CA, USA. [4]Molecular Biology Institute, University of California, Los Angeles, Los Angeles, CA, USA. [5]Eli and Edythe Broad Center of Regenerative Medicine and Stem Cell Research at University of California, Los Angeles, Los Angeles, CA, USA. [6]Jonsson Comprehensive Cancer Center, University of California, Los Angeles, Los Angeles, CA, USA. [7]These authors contributed equally: Jennifer Zou, Emily Maciejewski. ✉e-mail: jason.ernst@ucla.edu

approach has been previously applied to data from multiple marks across multiple cell types[23], but has not been applied in the context of multiple individuals in a single cell type.

One biological reason we could expect to observe recurring epigenetic patterns across individuals is that a TF may have differential activity across individuals[24–27]. This could be reflected in corresponding differential activity of histone modifications at many of its binding locations across the genome. Such TFs could potentially act as trans-regulators, that is proteins affecting the expression of multiple genes in the genome locally or distally. Distal impacts can be on the same or across chromosomes from where the gene is encoded. Reflecting the importance of trans-regulation, it has been estimated that 60–75% of the heritability in gene expression is explained by distal effects[28–31]. However, the identification of trans-regulators is challenging. This is the case compared with cis-regulatory elements because of the much larger number of association tests that traditionally need to be performed, which results in lower statistical power[32]. The recurring patterns of global variation have the potential to be more informative for identifying trans-regulators in a given cell type.

In this study, we use a stacked chromatin state model to systematically learn global patterns of epigenetic variation across individuals and annotate the genome based on them[21–23]. The stacked chromatin state model is based on a multivariate hidden Markov model (HMM) that learns combinatorial and spatial patterns across multiple individuals of one or more marks that recur in many regions of the genome. While the model training for this stacked chromatin state model follows previous applications of ChromHMM, the model input and interpretation is different. The hidden states of the stacked model are learned on data from multiple individuals in one cell type and represent recurring patterns across individuals and potentially also marks, which we will refer to as global patterns. We first develop and use the global patterns in lymphoblastoid cell lines (LCLs). Then, we demonstrate how this framework can be applied to analyze histone modification data from autism spectrum disorder (ASD) cases and controls in prefrontal cortex tissue. Previous studies have identified numerous molecular features, including RNA expression, RNA splicing, and histone modifications, that differ between ASD cases and controls[4,33–37]. We show that global patterns are also associated with diagnosis status, reflecting the known associations with molecular features. We expect identifying global patterns of epigenetic variation will be a useful framework to study epigenetic and transcriptional regulation networks and complex phenotypes in other systems.

## Results
### Systematic genomic annotation of chromatin variation across individuals

We applied a stacked version of the ChromHMM framework to learn a ChromHMM model where all histone modifications in all individuals are used as features[21–23]. We used genome-wide histone modification data quantified in 200 bp non-overlapping bins across multiple individuals and marks. We first regressed out the effects of known confounders before training the model (Methods). Similar to the ChromHMM framework when data from only a single cell type is used to train a model, we binarized the data using a Poisson background model and used this as input to ChromHMM[20,21]. Unlike the standard use of ChromHMM trained on a single individual, in this framework, each hidden state learned corresponds to a combinatorial and spatial pattern across individuals and potentially also marks, which we call a global pattern. Each global pattern has emission probabilities corresponding to the probability of observing a mark in a specific individual given the global pattern, which we use for downstream analysis. After the global patterns are learned, we annotate the genome at 200 bp resolution with the most likely hidden state of the HMM. This singular genome-wide annotation is universal to all individuals.

### Learning global patterns in LCLs

We first applied the stacked ChromHMM model to a data set of 75 individuals with three marks (H3K27ac, H3K4me1, and H3K4me3) (Supplementary Data 1) in LCLs[15]. We used this LCL data to showcase the utility of global patterns in a cell type that has previously been used to study histone

modification variation across individuals[5,15,38] with data for three different histone modifications that have previously been shown to be associated with different types of regulatory elements were available[12,15,39,40]. We trained stacked ChromHMM models to learn global patterns across both individuals and marks (Methods, Fig. 1). We learned models with between five and 100 states in increments of five and subsequently used these models to segment and annotate the genome according to these global patterns. We visualized the genome segmentation and annotation into global patterns of the final model alongside the histone modification data across individuals in custom genome browser tracks[41].

As an initial validation of the models, we tested whether the global patterns were internally consistent across pairs of the three histone modifications. As pairs of these three marks are known to co-occur in the genome[12,39], we expect the emission parameters for pairs of marks to be correlated across individuals in many of the states. As the patterns are from models learned agnostic to mark labels during training, correlation between pairs would support that the patterns are capturing reproducible cross-individual signal variation. To investigate this correlation, we performed an analysis to obtain the median Spearman correlation of emission parameters for each pair of histone modifications (Methods). The median pairwise correlation between marks increased rapidly until the number of global patterns was increased to 35. Correlations between pairs of histone modifications corresponding to active promoters (H3K4me3 and H3K27ac) and enhancers (H3K4me1 and H3K27ac) remained high (>0.5) for models with up to 100 states (Supplementary Fig. 1, Supplementary Data 1). High correlations were observed even though the models were learned agnostic to the mark and individual labels. This suggests that a global pattern is less likely to be caused by technical issues with the ChIP-seq experiments and more likely to be associated with differences at the sample level.

### Common genetic variation associated with LCL global patterns

To investigate the genetic basis of global epigenetic patterns across individuals, we performed what we termed global pattern quantitative trait association analysis. For each model that we had learned, we tested association of common variants in the data set[15] with the emission parameters of each global pattern to identify significantly associated ($p_{adj} < 0.05$) global pattern quantitative trait loci (gQTLs). To maximize the number of discovered gQTLs we focused our analysis on the model with the number of hidden states that maximized this quantity (Methods). Specifically, the number of gQTLs was maximized with the 85-state model (2945 gQTLs, Supplementary Fig. 2, Supplementary Data 2). In this 85-state model, 36 states were associated with at least one gQTL. We verified that global patterns for the 85-state model were robustly learned across different subsets of the genome (Median Spearman correlation of emissions = 0.93, Methods, Supplementary Fig. 3). We additionally performed a replication analysis for these identified gQTLs using data from the BLUEPRINT consortium[42]. The $p$ values obtained in the replication analysis were more significant than expected by chance (Mann–Whitney $U$ Test (two-sided) $p = 0.03$, Methods, Supplementary Fig. 4, Supplementary Data 2), supporting that the gQTLs identified using the LCL global patterns contain reproducible signal.

We then sought to understand the biological relevance of the gQTLs identified to the LCL cell type by performing a gene set enrichment analysis for the gQTLs using GREAT[43]. GREAT analyzes genes near a set of genomic regions and tests for ontology and phenotype enrichments compared to a background set of regions. The "regulation of lymphocyte activation" gene ontology (GO) term was significantly enriched for the gQTLs (FDR < 5%, Supplementary Data 2). This enrichment was expected since LCLs are derived from lymphocyte cells, which are critical for immune system functions. While we also observed significant enrichment for terms not directly related to immune function, we did not expect all global patterns to associate with cell type-specific processes. Additionally, there may be pleiotropy between immune function and other complex traits. Overall, enrichment of terms relevant to immune function suggest that some gQTLs are biologically relevant for the LCL cell type and that global patterns may be informative for identifying sources of molecular variation associated with immune function.

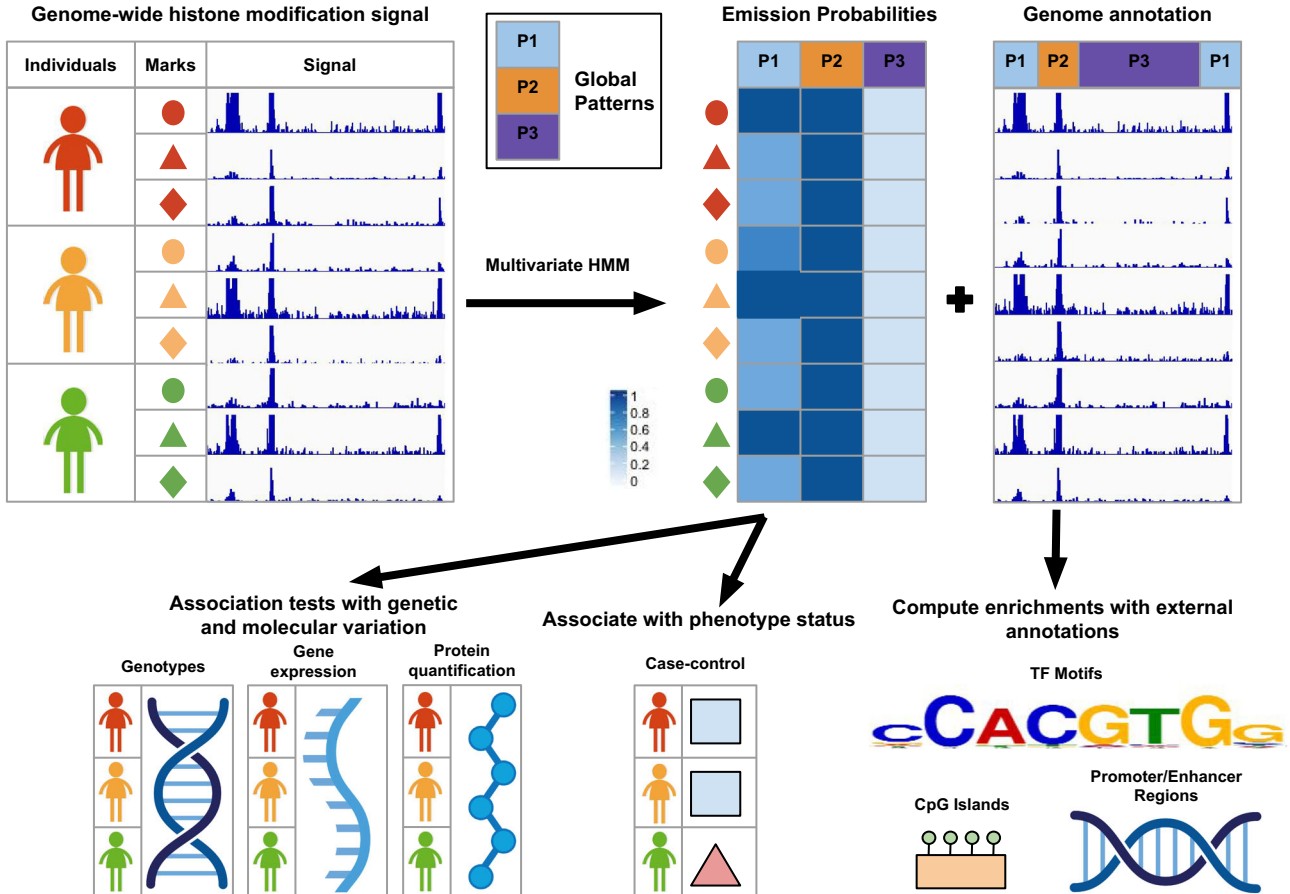

**Fig. 1 | Method overview.** The figure provides an overview of our methodological approaches. We first trained a stacked multivariate HMM model using genome-wide histone modification signal from multiple individuals and for one data set also marks ("Genome-wide histone modification signal") using ChromHMM. This model learns global patterns of epigenetic variation that recur in many regions of the genome ("Emission Probabilities"). The model learning is agnostic to the mark or individual label of each data set. The emission probabilities learned for a hidden state correspond to the probability of observing a presence call for each data set conditioned on being in the hidden state. We used the model learned to annotate the genome according to these patterns at 200 bp resolution ("Genome annotation"). We then used both the model emission parameters and global pattern annotations for various downstream analyses, including but not limited to the examples illustrated in the figure. We associated the emission probabilities of the global patterns with other measures of molecular variation or phenotypic status. Additionally we computed for locations annotated to specific global patterns enrichments for external annotations. We generated the example motif image with TFmotifView[64].

We further explored the gQTLs identified in the LCL cell type by conducting an eQTL analysis between the genotype information and external gene expression[15] in the subset of individuals ($n = 54$, Supplementary Data 1) for which gene expression data was available. We associated both the full set of gQTLs with all available genes and the subset of gQTLs with MAF > 5% considering the subset of individuals with gene expression data available with all available genes. We did the same for random permutations of SNPs and identified the set of significant eQTLs (Bonferonni corrected $p$ value < 0.05) when considering only the top-most associated SNP for each unique gene for the eQTL set (Supplementary Data 2). We observed that the number of significant eQTLs from this filtered set found in the gQTL set was greater than 96% of the number of eQTLs found in any of the random SNP permutations when considering the full set of gQTLs. The number of overlapping significant eQTLs was greater than the number in 90% of permutations when considering the subset of common variant gQTLs based on the subset of individuals with gene expression data available at MAF > 5% (Methods). These results suggest that gQTLs may contain signal related to gene expression and help identify cis- and trans-regulatory effects.

**LCL global patterns enriched for active regions of the genome**
To characterize the types of genomic regions found in the global patterns learned in the 85-state model (Supplementary Data 3), we computed enrichments for previous reference chromatin state annotations for a LCL cell line (Methods, Supplementary Fig. 5, Supplementary Data 4). These chromatin state annotations correspond to combinatorial and spatial patterns of LCL epigenetic data sets for a single individual. We identified the chromatin state with the highest fold enrichment among states significantly enriched (overlap enrichment > 1 and Binomial Test (two-sided), FDR < 5%; Methods) for each global pattern. The majority of global patterns were most highly enriched for enhancer and promoter chromatin states. We defined these global patterns as enhancer- and promoter-like global patterns, respectively. The high correspondence with enhancer and promoter chromatin states was expected given the histone modifications used to learn the model are known to be associated with enhancer and promoter activity (Fig. 2A) and supports the relevance of the annotations for analysis of epigenetic variation at regulatory regions.

We also evaluated the enrichment of global patterns for external annotations that were not based on histone modifications. These included CpG Islands[44], DNase I hypersensitive sites[15], and promoter regions defined as positions within 2kb from GENCODE annotations transcription start sites (TSS)[45], which as expected showed significant enrichments for many global patterns (Fig. 2B, Binomial Test (two-sided), FDR < 5%, Supplementary Data 4). CpG Islands tended to be enriched in promoter-like global patterns, whereas DNase I hypersensitive sites are enriched in both promoter-like and enhancer-like global

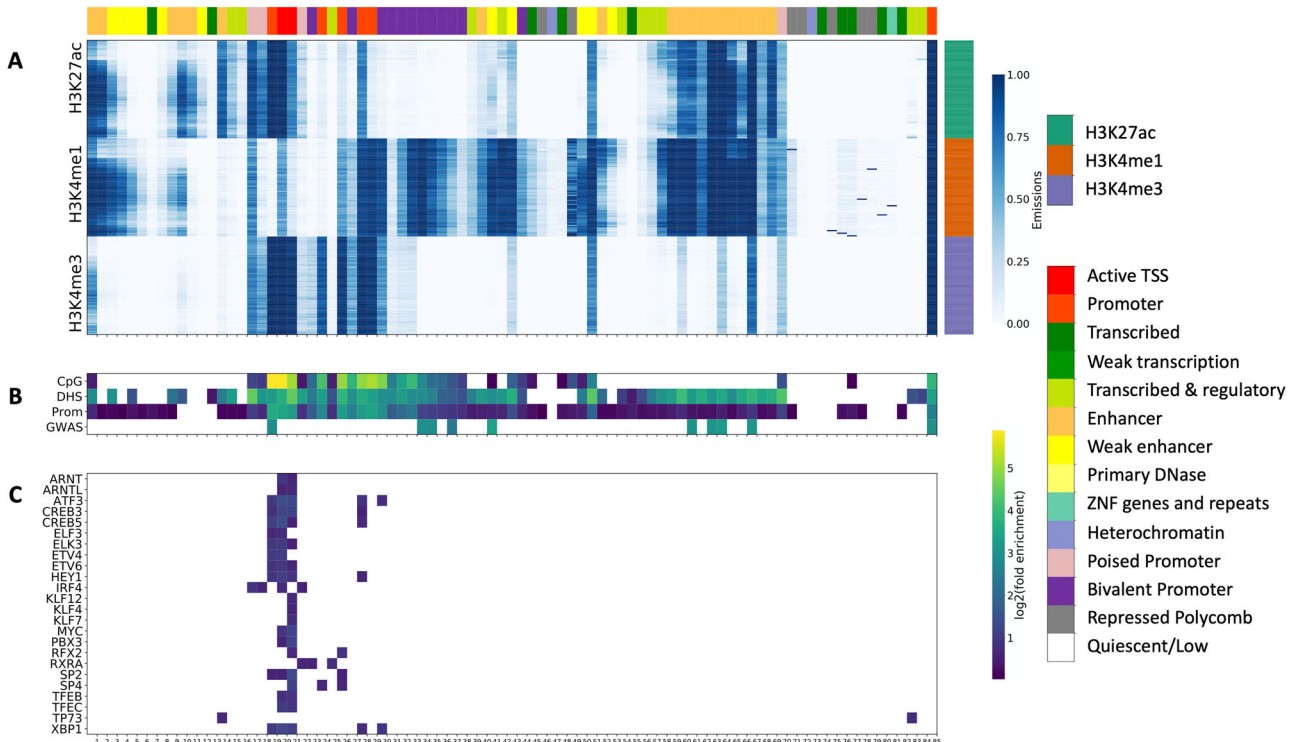

**Fig. 2 | LCL 85-state model. A** Emission matrix for the 85-state LCL model. The x-axis ("Global Pattern") shows the global patterns learned, and the y-axis shows the data sets, which are grouped by the indicated histone modifications. The ordering of individuals is the same for each histone modification. Each global pattern is annotated (color bar across top of heatmap) with a previous LCL chromatin state annotation for one individual[60] with the highest fold enrichment, labeled by the color bar on the right. **B** Overlap enrichments for CpG Islands[44], consensus DNase I

hypersensitive sites[15] measured in the same samples, promoter annotations computed from GENCODE transcription start site (TSS) annotations[45], and immune-related GWAS variants[9]. Only significant enrichments are shown (Binomial Test, FDR < 5%, fold > 1). **C** Motif enrichments for 24 TFs with motifs strongly enriched in at least one global pattern (FDR < 5%, log2(fold enrichment) > 1.5) and gene expression associated with global patterns (FDR < 5%).

patterns. We also evaluated enrichments for fine-mapped GWAS variants for 39 immune-related diseases[9], which revealed nine global patterns of histone modifications with significant enrichments (Fig. 2B, Binomial Test (two-sided), FDR < 5%). These GWAS enriched global patterns were strongly enriched for promoters and enhancers, which is consistent with previous analyses[9,12,46].

We additionally investigated to what extent the data from subsets of the three histone modifications are sufficient to recover the original genome annotation from the 85-state model trained on all three marks (Methods). We found that annotations based on decoding with a single mark had only moderate ability to recover the original genome annotation based on the full set of marks, especially for enhancer and promoter-like patterns. Regions of the genome where the original annotation was not recovered were primarily annotated into global patterns enriched for heterochromatin and transcription associated chromatin states (Supplementary Fig. 6, Supplementary Data 1). Using any two marks increased the proportion of recovery, particularly in active TSS, promoter, and enhancer-like patterns, but still did not allow for full recovery of the annotation based on the full set of marks (Supplementary Fig. 7, Supplementary Data 1).

Overall, the learned LCL global patterns were highly enriched for promoter and enhancer-like patterns which agreed with external annotations and were informed by patterns of variation across the three LCL histone modifications.

**LCL global patterns associated with large-scale molecular variation**

In addition to using external annotations of the human genome, we associated the global patterns corresponding to each mark with external gene

expression[15] and protein quantification[38] data sets across individuals, which were not used to train the models. We conducted the association with the expression of genes in a subset of the individuals (*n* = 54, Supplementary Data 1) for which gene expression data was available. We identified 4774 genes with expression patterns significantly associated with at least one global pattern (Linear model, FDR < 5%, Supplementary Data 5), indicating that the global patterns co-vary with the expression of a large number of genes. We also computed the average correlation between each global pattern for each mark and gene expression at different distances to the global pattern in the annotation (for each 100 bp window, up to 500 kb upstream and downstream, Methods). We found that global patterns tended to more highly associate with genes in relatively closer proximity (i.e, 100 kb, Supplementary Figs. 8–10, Supplementary Data 5), as expected.

We additionally tested for association of the global patterns corresponding to each mark with co-expression modules generated from the subset of individuals for which gene expression was available, which represent patterns in gene expression across individuals (Methods). The association test was performed separately for each mark. This revealed that the module eigengene values are highly concordant with the global patterns with 16 of 19 modules significantly associated with at least one global pattern and 72 global patterns represented among these associations (Linear model, FDR < 5%, Supplementary Fig. 11, Supplementary Data 5). Among global patterns that have a significant association with at least one co-expression module, 62.5% were most associated with enhancer and promoter chromatin states. The remaining global patterns not associated with a module tend to have lower signal and be most associated with repressed polycomb, heterochromatin, and transcribed states (Fig. 2A, Supplementary Fig. 11). Additionally, certain groups of global patterns tended to be associated with

the same co-expression module consistent with inter-pattern correlation. Specifically, pairs of global patterns that are associated with the same co-expression module tend to have substantially higher pairwise correlation between emission parameters than pairs of patterns not significantly associated with the same module (Mann–Whitney $U$ Test $p$ value < 0.001, Supplementary Fig. 12, Supplementary Data 5).

Certain co-expression modules exhibited strong association with multiple patterns while other modules exhibited few to no associations with any global pattern. The difference in behavior between modules is associated with the genes assigned to each module. Based on the previous analysis, approximately one-third of the genes considered in this study are significantly associated with at least one global pattern, with all 85 global patterns included among these associations and certain genes being associated with multiple global patterns. Co-expression modules with a higher percentage of genes included in this subset of significant genes are more likely to be associated with multiple global patterns Supplementary Fig. 13). Moreover, the strength of the association between co-expression modules and multiple global patterns is associated with the number of genes associated with a global pattern within a co-expression module (Supplementary Fig. 14–17, Supplementary Data 5). This relationship between global patterns and co-expression modules being associated with the behavior of specific genes within each module is expected, as a co-expression module captures shared expression patterns across genes. The relationship between co-expression modules and the genes they represent along with the existence of a substantial number of genes significantly associated with at least one global pattern is consistent with that certain modules are strongly associated with global patterns. Conversely, the existence of a substantial number of genes not associated with any global patterns is consistent with other modules not being associated with any global patterns. Overall, these results from both the gene expression and co-expression module association tests demonstrate that global patterns of epigenetic variation are associated with large-scale expression patterns, despite expression data not being used during model training.

We also associated the global patterns with protein abundance data for a subset of 60 out of the 75 individuals for which the protein data was available[38]. Of the 60 individuals with both protein abundance data and histone modification data, 44 also had gene expression data (Supplementary Data 1). After intersection of the protein data with their corresponding gene expression data[15], there were 4319 genes with protein abundance data and gene expression data. Of the 4774 genes with expression patterns significantly enriched for at least one global pattern, 1349 also had protein data available. We correlated the abundance of each protein across the 60 individuals with the emission parameters for each global pattern (Methods). We identified 594 proteins that were significantly associated with at least one global pattern (Linear model, FDR < 5%), 588 of which had gene expression available. Of these 588 proteins, we identified 258 proteins that also had differential expression significantly associated with at least one global pattern, which was a significant overlap (Fisher's Exact test $p$ value < 0.001, Fold Enrichment 1.4, Supplementary Fig. 18, Supplementary Data 5). Of these 258 proteins, 143 were associated with at least one of the same global patterns as its corresponding gene expression values. We note the partial agreement with corresponding differential gene expression could be explained by a number of factors. First, of the 4774 genes with expression patterns significantly associated with a global pattern, 3425 did not have protein data available. Second, the number of samples differed between the data sets with only 44 samples shared between protein abundance, histone modifications, and gene expression. Since global patterns reflect different subsets of individuals with histone modification signal, lack of overlap could account for the partial agreement. Finally, proteins with differential expression but not differential protein levels could potentially be explained by post-transcriptional differences between individuals[38,47].

Overall, LCL global patterns based on epigenetic variation across individuals were associated with large-scale molecular variation across individuals as determined by other assays not used during model training, further supporting their biological relevance.

## Predicting trans-regulators in LCL

As the global patterns of histone modifications might be associated with TFs that have differential activity across individuals, the identification of enriched TF motifs at genomic positions annotated to specific patterns may suggest trans-regulators associated with them.

Thus, in order to identify potential trans-regulators, we first performed TF motif enrichment analysis for each global pattern. We calculated motif enrichment of 602 TF motifs in the ENCODE motifs database[48] compared to shuffled motifs. We observed strong enrichment of motifs for 20 of the 85 global patterns, corresponding to 79 distinct TFs (Fisher's Exact test, FDR < 5%, $log_2$(fold enrichment) > 1.5), with the number of enrichments increasing to 42 global patterns when we relaxed the enrichment requirements to $log_2$(fold enrichment) > 1. These TF motifs were primarily enriched in promoter-like global patterns, with some corresponding to enhancer-like global patterns (Fig. 2A, Supplementary Data 4). This motif enrichment in putative regulatory regions of the genome provides orthogonal support of the likely regulatory nature of these genomic regions.

To provide additional evidence that some of the TFs corresponding to these motifs potentially have trans-regulatory activity we analyzed their correlations with gene expression. In total, 24 of the 79 TFs with motifs enriched in a global pattern, also have their expression levels significantly associated with a global pattern (FDR < 5%, Fig. 2C, Supplementary Fig. 19, Supplementary Data 4), although not necessarily the same pattern. We highlight two TFs (TP73 and CREB5) which had gene expression patterns associated with the same global patterns for which their motifs were enriched. These genes provide examples of how global patterns can be used to identify potential trans-regulators supported by multiple types of evidence, but we note many other genes have their expression levels significantly associated with the global patterns.

TP73 has been shown to regulate $T$ helper differentiation-related genes, which results in variation in autoimmune disease susceptibility in mice[49]. Additionally, the TP73 motif was associated with an enhancer-like global pattern that is also significantly associated with TP73 gene expression (Supplementary Fig. 20). We show an example instance of the TP73 motif located in a region assigned to this global pattern in Supplementary Fig. 21. The CREB family of TF regulate genes containing a cAMP-responsive element, including a number of immune-related genes[50]. We found the CREB5 motif and its gene expression are associated with global patterns associated with active TSS regions (Supplementary Figs. 22, 23).

TFs with expression associated with a global pattern, particularly the same one in which they have an enriched motif, have additional evidence of trans-regulatory activity. We note that TFs might be post-transcriptionally regulated, which could lead to motif enrichments in specific global patterns without corresponding gene expression correlations. Overall, TF motif enrichment analysis for global patterns and in some cases combined with gene expression associations can assist in the identification of potential trans-regulators.

## Learning global patterns across ASD cases and controls

To investigate global patterns in the context of a complex disorder, we focused on ASD. Various molecular features including the histone modification H3K27ac have been mapped in ASD case along with control individuals[4]. Specifically, we learned a separate model using H3K27ac histone modification data from prefrontal cortex tissue in ASD cases and controls[4].

We used the same procedures as the LCL data set to preprocess the data, train models, and select the number of hidden states except that we accounted for a larger set of known covariates than were used in previous differential peak analyses (Methods)[4]. We selected the 90-state model (Fig. 3) for follow-up analyses, which maximized the number of gQTLs (Supplementary Fig. 24, Supplementary Data 2) with 2229 gQTLs. Similar to the LCL analysis, we performed GREAT[43] enrichment using the gQTLs as the foreground and the whole genome as the background. We identified a number of enriched GO terms (FDR < 5%, Supplementary Data 2) relevant for the prefrontal cortex tissue, such as "neuron projection morphogenesis,"

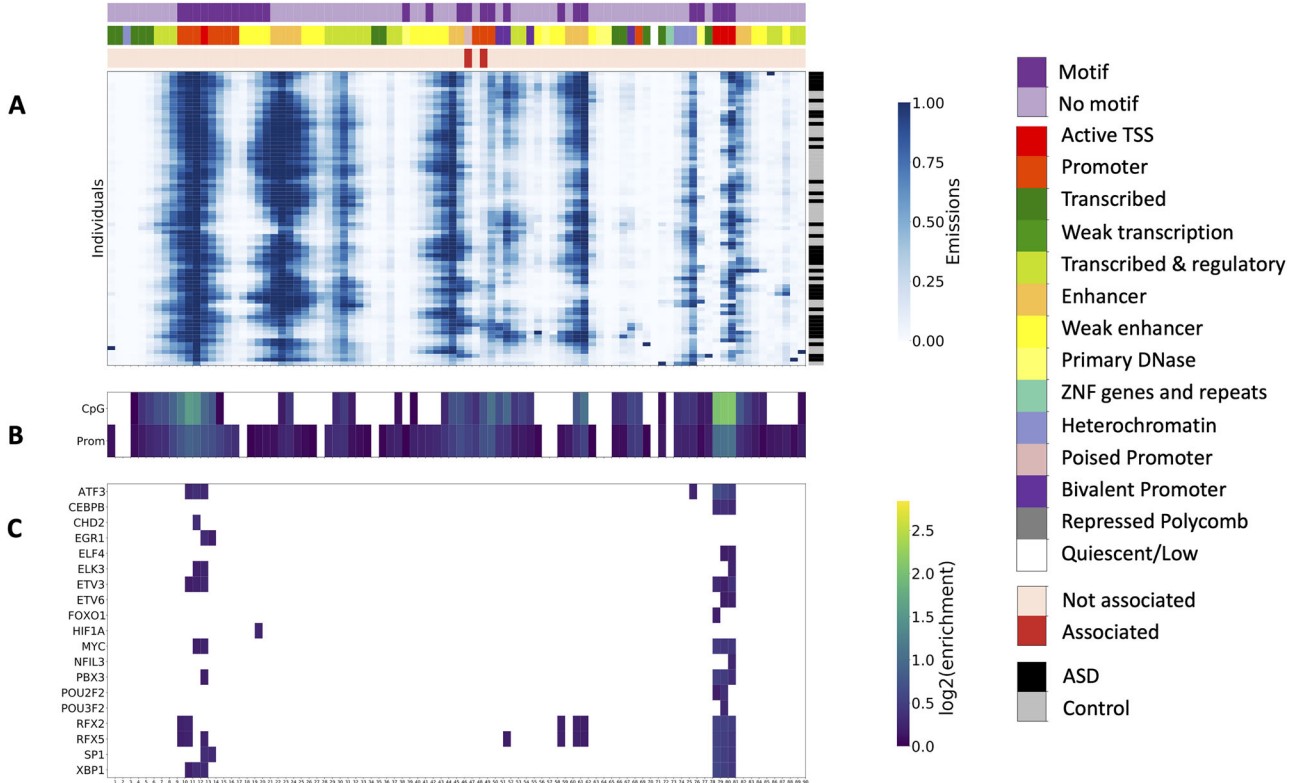

**Fig. 3 | ASD 90-state model. A** Emission matrix for the 90-state ASD model. The *x*-axis ("Global Pattern") shows the global patterns learned, and the y-axis shows each individual ("Individuals"). The diagnosis status of each individual is annotated (color bar on right). Each global pattern is annotated in the top three rows with whether the global pattern had significant motif enrichments (first), the reference chromatin state annotation for prefrontal cortex tissue[60] with the highest enrichment (second), and whether the state was significantly associated with diagnosis status (third). Color legends for each color bar can be found on the right of the heatmap in the same order from top to bottom. **B** Overlap enrichments for CpG Islands[44] and promoter annotations computed from GENCODE TSS annotations[45]. **C** Motif enrichments for 19 TFs with motifs strongly enriched in at least one global pattern (FDR < 5%, log2(fold enrichment) > 1.5) and gene expression associated with global patterns (FDR < 5%).

"axon development", and "regulation of axon guidance". While we also observed significant enrichment for terms not directly related to brain function, we did not expect all global patterns to be associated with cell type-specific processes.

We next computed enrichments of global patterns for chromatin states previously annotated in the prefrontal cortex tissue from the Roadmap Epigenomics Consortium[51] (Supplementary Fig. 25, Supplementary Data 4). We annotated each global pattern with the most enriched state from this single reference epigenome annotation (Fig. 3A, Supplementary Fig. 25). Like the LCL model, most of the states in the ASD model were highly enriched for promoter and enhancer chromatin states (Binomial test (two-sided), FDR < 5%) (Fig. 3A). Consistent with the LCL data set, we also found in promoter-like global patterns high enrichments for CpG Islands and bases within 2 kb of GENCODE annotated TSS (Fig. 3B, Supplementary Data 4).

We identified 3535 genes with expression associated with the ASD global patterns (FDR < 5%, Supplementary Data 5). To identify potential trans-regulators, we computed TF motif enrichments for each global pattern. As with the LCL data, we calculated motif enrichments of 602 TF motifs in the ENCODE motifs database[48] compared to shuffled motifs. We observed strong enrichment of motifs for 27 of the 90 global patterns, corresponding to 114 distinct TFs (Fisher's Exact test, FDR < 5%, *log2*(fold enrichment) > 1.5, Supplementary Data 4). In total, 19 of the 114 TFs with motifs enriched in a global pattern, also had their expression levels associated with a pattern (FDR < 5%, Fig. 3C, Supplementary Data 4), although not necessarily the same pattern. For example, we found the RFX family's motif, which was also previously identified to be enriched in differential peaks between cases and controls[4]. Furthermore, it was noted that RFX2

contains a differentially acetylated peak in its promoter, suggesting a potential mechanism by which its expression differs across individuals[4].

As the ASD data set contains both case and control individuals, we associated the global patterns with the diagnosis status. This identified two global patterns (47 and 49) that were significantly associated with ASD status (two-sided Mann–Whitney $U$ $p_{adj}$ < 0.05, Methods, Fig. 4, Supplementary Figs. 26, 27, Supplementary Data 3, 5). In addition to the Mann–Whitney $U$ test, we performed logistic regression between global patterns 47 and 49 including covariates and the case-control status (Methods). We found global patterns 47 and 49 to still be associated with ASD status when directly accounting for covariates (two-sided $z$-test $p_{adj}$ < 0.05). This was expected as the covariate correction performed on the histone signal used to learn the global patterns removed almost all correlation between signal and covariates (Supplementary Fig. 28, Supplementary Data 1). Both global patterns were highly enriched for promoter regions of the genome. Global pattern 47 was significantly associated with the expression of 22 genes, and global pattern 49 was significantly associated with the expression of 316 genes (FDR < 5%). The genes associated with state 49 were enriched for several GO terms (Supplementary Table 1), including "cellular response to interleukin-1," which has previously been suggested to play a role in the disorder etiology[52]. These results suggest that analyzing global patterns can potentially be informative towards studying complex phenotypes.

## Discussion

In this work, we learned global patterns of epigenetic variation across individuals and systematically annotated the human genome according to these patterns of variation using stacked ChromHMM chromatin state

**Fig. 4 | Association with ASD diagnosis status.**
**A**, **B** Plots corresponding to a global pattern significantly associated with the ASD phenotype. The diagnosis status is shown on the *x*-axis and the emission parameters for the state are shown on the *y*-axis. In both plots, the center line of the boxplot corresponds to the median emission across samples. The box limits correspond to the upper and lower quartiles. We tested for association between emission parameters for each state in the 90-state ASD model and phenotype status. After controlling for multiple testing using a permutation test, the two global patterns shown, **A** global pattern 47 and **B** global pattern 49, were significantly associated with diagnosis status ($p_{adj} < 0.05$, Mann–Whitney *U* Test).

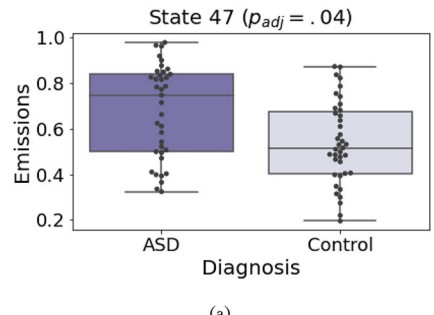
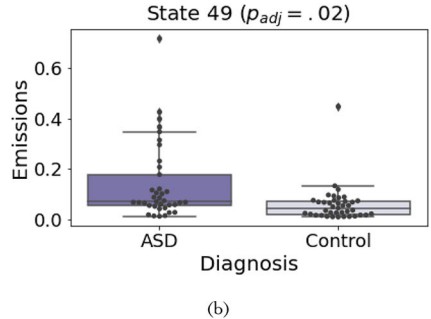

models. We applied this framework to an LCL data set for three histone modifications (H3K27ac, H3K4me1, and H3K4me3) and an ASD case and control data set for one histone modification (H3K27ac).

Previous work in detecting molecular variation across individuals have performed marginal association tests on consensus regions of the genome[1–5]. Individual specific chromatin states were previously defined[3], but these did not directly identify patterns of epigenetic variation across individuals that recur in the genome as the stacked modeling does. Another potential advantage of the stacked modeling approach is that in general it should be less sensitive to missing epigenetic marks in an individual compared to approaches that define individual specific chromatin state annotations.

We identified motifs of TFs enriched in the global patterns for both the LCL and ASD data enabling us to predict potential trans-regulators. Of these TFs, a fraction also had differential expression associated with at least one global pattern. In these cases, there is additional evidence of molecular coordination in different regions of the genome. We also identified thousands of genes in both data sets with expression patterns significantly associated with the global patterns.

One challenge in identifying global patterns is that it is difficult to distinguish global patterns due to confounders and those due to biological reasons, such as trans-regulation. Unsupervised methods, such as principal component analysis and PEER[53] are likely to remove true biological signal in these applications. To mitigate the effect of confounders, we regressed out known covariates from the histone modification input signal. In the LCL data set, we showed that the biological signal of interest was consistent across multiple histone modifications. To further demonstrate the likely biological significance of the global patterns we used external data and annotations. For both the LCL and ASD data sets, we identified a number of global patterns that showed consistent co-variation with gene expression. We also showed that the global patterns were associated with common genetic variants and that these variants were in close proximity to tissue/condition relevant genes. These analyses support that the differences across individuals captured is biological. However, we cannot exclude that technical differences are also driven by sample-level confounding factors that were not previously regressed out.

Finally, we identified two global patterns significantly associated with ASD, in spite of the limited sample size and heterogenous nature of ASD. The framework we have used is general and can be applied or adapted to other data sets from different cell types, phenotypes, and other types of epigenetic data such as DNA methylation or chromatin accessibility. These potential extensions would allow for further analysis and understanding of epigenetic variation across individuals and its relationship with phenotypes.

## Methods
### Histone modification data
We learned global patterns for LCL histone modification data[15] and ASD histone modification data[4]. For the LCL data, histone modification signal was mapped for three marks (H3K27ac, H3K4me1, and H3K4me3) in 75 individuals using procedures described in ref. 15. Briefly, reads were aligned to personal genomes and filtered for unmapped reads, duplicates, unpaired reads, and reads with low quality. The BAM files from that work were used for this paper. We did not perform any additional quality control beyond what was previously applied. For the ASD data, H3K27ac signal was mapped in the prefrontal cortex tissue for 93 individuals. Reads were mapped to the human reference genome and duplicate reads were filtered. Extensive sample quality control was performed in ref. 4, resulting in a number of samples excluded. We used BAM files for the same 76 individuals that were used in previous work after quality control[4]. For this data, we did not perform any additional quality control beyond what was previously applied. We only included autosomes in our analysis and we used the hg19 genome assembly.

### Learning epigenetic patterns across individuals using a stacked ChromHMM model
We quantified the number of reads falling within each 200 bp non-overlapping bin in the genome using the BinarizeBam command in the ChromHMM software package (version 1.11)[21]. The approach of using non-overlapping bins of 200 bp length, corresponding to roughly the size of a nucleosome and flanking region, follows the default for ChromHMM and what has been effectively applied in many prior applications of ChromHMM. For each bin, we then fit a quasi-poisson regression model, where the dependent variable was the number of reads in the bin and the independent variables were standardized known covariates[54]. Quasi-poisson regression handles both over and under-dispersion of count data and has been used in related applications[55]. We first quantile normalized the read counts so that the distribution of counts for each individual across bins in the genome was the mean value of the quantile. For each data set, we then regressed out the effect of the known covariates that were used in previous work. For the LCL data set, we accounted for sex, genotyping method, number of reads and relative strand correlation (RSC)[15]. We performed the correction separately for each mark in the LCL data set. For the ASD data set, we accounted for age, sex, percentage of neuronal cells, brain bank, number of peaks, fraction of reads in peak (FRIP), read duplicate fraction, and read alignment fraction[4]. The percentage of neuronal cells was previously determined using the CETS algorithm and DNA methylation data collected in the same cohort[56]. We then binarized the corrected histone modification count data according to the procedures used in ChromHMM using the BinarizeSignal command with default parameters[21]. We used binarized read count data as input to learn the parameters to a stacked ChromHMM model. To learn the model parameters and perform genome segmentation and annotation we used the LearnModel command of the ChromHMM software package with default parameters[21]. We trained stacked models where each ChIP-seq data set was treated as a separate mark. We trained separate models using 5–100 hidden states in increments of five states for each data set.

## Validation of learned global patterns

For each LCL model trained with between 5 and 100 hidden states, we evaluated the pairwise correlations of emission patterns between marks for the same pattern across individuals. For each pair of histone modifications (H3K27ac-H3K4me1, H3K27ac-H3K4me3, and H3K4me1-H3K4me3), we performed the following analysis to obtain the median Spearman correlation of emission parameters. For each global pattern learned by the model, we first computed the Spearman correlation between the emission parameters for the first mark in the pair with the emission parameters for the second mark. After computing this correlation for each global pattern, we took the median Spearman correlation across all global patterns for the model.

## gQTL analysis

We associated each global pattern with common variants (MAF > 5%) previously identified in each respective data set[4,15]. In these association analyses, we split the emission matrix by mark and treated the emission parameters for each mark and state as a phenotype. We performed the associations and computed the $p$ values for each association using the MatrixEqtl v2.2 software[57]. For the LCL data, we included the first 20 genotyping principal components (PCs) as covariates when performing the associations. The PCs were calculated using the Plink software version v1.90b6.24[58]. We used a Bonferonni corrected threshold of $\frac{5*10^{-8}}{mk}$, where $m$ in the number of marks, and $k$ is the number of global patterns. For the LCL data set, we performed the gQTL analysis on all 75 individuals with Yoruba ancestry. For the ASD data set, we performed the gQTL analysis using only individuals with Caucasian ancestry (69/76 individuals).

## Choosing the number of hidden states

To select the number of hidden states, which corresponded to the number of global patterns, we analyzed models that we had trained using between 5 and 100 hidden states in increments of five states. We then associated common variants (MAF > 5%) in the human genome with the emission parameters to identify gQTLs and chose the number of hidden states that maximized the number of gQTLs.

## Evaluating robustness of global patterns

To evaluate whether the global patterns are robustly learned across different subsets of data, for each number of states between 5 and 100, we first trained two models using chr1 and chr2 separately. For each pair of models with the same number of states, we matched states from the two models using a greedy approach. We iteratively computed pairwise Spearman correlations between unpaired states, matching the two states from the models with the highest correlation.

## gQTL replication

We performed a replication analysis on the gQTLs identified in the LCL data set. We first learned global patterns in BLUEPRINT histone modification data[42]. The histone modification signal was for two marks, H3K27ac and H3K4me1, for 154 and 152 individuals, respectively, from mature neutrophil cells[42]. We trained stacked ChromHMM models, identified gQTLs, and selected the number of hidden states using the same methodology as for the LCL data. We used the 80-state model from the BLUEPRINT data for replication as it maximized the number of gQTLs.

We found the BLUEPRINT data to have a greater number of what we refer to as singleton global patterns, where the global pattern corresponds to high histone signal in exactly one individual, compared to the YRI LCL data. To increase power in the replication analysis, we restricted the set of SNPs considered for gQTLs in the BLUEPRINT data to those that were associated with nonsingleton global patterns. This is analogous to filtering out SNPs with lower minor allele frequency. We identified nonsingleton global patterns as follows. For each global pattern, we counted the number of individuals with emission parameters greater than 0.5 within that global pattern. We defined singleton states as those with only one individual with emission parameter greater than 0.5. All other states were classified as nonsingleton

states, including states where all individuals had emissions less than 0.5. Of the 80 global patterns in the final BLUEPRINT model, we classified 54 as nonsingleton states.

Of the 2945 SNPs identified as significant gQTLs in the LCL data set, 739 were available in the BLUEPRINT data set. We attempted to replicate each of these SNPs in the BLUEPRINT data set by associating each with the 54 nonsingleton global patterns. We performed this association test using the same methodology as the LCL gQTL analysis and included the first 20 genotyping PCs as covariates, calculated using the Plink software version v1.90b6.24[58]. We compared the $p$ values obtained in the BLUEPRINT replication with a uniform distribution using a two-sided Mann–Whitney $U$ Test statistical test.

## GREAT enrichment analysis

We performed GREAT enrichment[43] for the set of gQTLs identified by each data set, using the gQTLs as the foreground and the whole genome as the background. The whole genome was selected as the background distribution because variants from all regions of the genome were tested for gQTL association. Using default parameters, GREAT computes a basal regulatory domain of a minimum distance of 5 kb upstream and 1 kb downstream. The regulatory domain is extended in both directions to the nearest gene's basal domain, but no more than 1000 kb in one direction. gQTLs are intersected with these regulatory domains to make gene assignments for the gQTLs. GREAT uses a hypergeometric test to assess whether the genomic regions in the foreground are significantly enriched for GO terms relative to the background. We identified GO terms that were significantly enriched in the gQTLs (FDR < 5% and observed genes > 5). We additionally performed a GO enrichment analysis on the 316 genes associated with global pattern 49 in the ASD model using the whole genome as the background.

## Overlap of gQTLs and eQTLs

For the LCL data set, we used the available genotypes and external RNA-seq data to perform an eQTL analysis and determine the overlap between gQTLs and the identified eQTLs. Expression data was previously collected for 54 of the 75 samples using RNA-seq[15] (Supplementary Data 1). We corrected the expression data for known covariates (sex, genotyping method, number of reads, the ratio between the fragment-length peak and the read-length peak or RSC[59]). We considered the same set of genes as in a previous publication[15].

To compute eQTLs, we computed the allele frequency of each gQTL both considering all available individuals and only considering the subset of 54 individuals with gene expression data available. We identified a subset of 1016 gQTLs with MAF > 5% based on this subset of 54 individuals. For both the full gQTL set and subset of gQTLs, we generated 100 random permutations of the same size and allele frequency distribution as the gQTL set. We performed an association test between the gQTL sets plus the corresponding 100 random permutations for each set and the gene expression data to identify eQTLs. We calculated the association statistics for a linear model using the MatrixEqtl v2.2 software[57] and used a Bonferonni corrected $p$ value of 0.05 to identify significant eQTLs. We considered the top-most significant SNP for each unique gene in the eQTL set. The number of gQTLs that resulted in a significant eQTL by this threshold was compared to the number of significant eQTLs identified in each random SNP permutation.

## Overlap enrichments with external annotations

We computed the fold enrichment for global pattern annotations overlapping external annotations in both the LCL and ASD models using the OverlapEnrichment command in the ChromHMM software (Version 1.11)[21]. To assess the significance of the overlap, we computed significance $p$ values for the enrichments using a two-sided binominal test, where the probability of success was set to the fraction of bases covered by the annotation in the genome. We corrected for multiple testing using an FDR threshold of 5% (Benjamini–Hochberg procedure) for all annotations tested within each model.

For both models, we computed enrichments for tissue-specific chromatin state annotations using a previous 25-state concatenated model that was learned from imputed data for 12-marks[60] based on data from the ENCODE Consortium Project[61] and the Roadmap Epigenomics Project[51]. For the LCL model, we used the lymphoblastoid cell line (reference epigenome E116), and for the ASD model, we used the prefrontal cortex tissue (reference epigenome E073). We tested enrichment for each of the 25 states in these models and annotated the global patterns with the state with the highest enrichment. We also computed enrichments for promoter annotations computed from GENCODE V29 TSS annotations[45], which we defined as regions 2 kb from the TSS of the genes. Finally, we computed enrichments for CpG Island annotations obtained from the UCSC Genome Browser[41].

For the LCL model, we computed enrichments for DNase I hypersensitive sites collected in the same individuals[15]. Specifically, we tested previously published consensus peaks that were obtained by merging peaks across individuals (http://mitra.stanford.edu/kundaje/portal/chromovar3d)[15]. We also obtained 4950 candidate causal SNPs for 21 immune diseases that were previously fine-mapped from 636 autoimmune GWAS loci[9] and tested them for enrichment in the global patterns.

### LCL annotation recovery with subsets of histone modifications

We evaluated the annotation recovery with subsets of marks from the LCL model using the EvalSubset command in the ChromHMM software (Version 1.11)[21]. This command evaluates the proportion of the annotations based on the full input that can be recovered when using a subset of the input for decoding. The output consists of a confusion matrix indicating the proportion of the genome annotated into each global pattern based on the full and subset of marks. The diagonal of the confusion matrix represents the proportion of the genome annotation assigned to the same pattern when using a subset of marks relative to when using all three marks for decoding.

### Association of global patterns with gene expression, co-expression modules, and protein quantification

For the association of global patterns with gene expression, we used external RNA-seq data, which was not used to train the model. Expression data was previously collected for 54 of the 75 LCL samples using RNA-seq[15] (Supplementary Data 1). We corrected the expression data for known covariates (sex, genotyping method, number of reads, the ratio between the fragment-length peak and the read-length peak or RSC[59]). We considered the same set of genes as in a previous publication[15]. Expression data was available for 51 of the 76 individuals from the ASD data set[34]. For this data set, we used expression values previously corrected for GC content and batch effects and subject outlier removal as described in ref. 34. We considered the same set of genes as in this previous work. We tested for association between gene expression and the emission parameters separately for each mark. We calculated the association statistics for a linear model using the MatrixEqtl v2.2 software[57] and used an FDR threshold of 5% within each mark.

Using the LCL global pattern genome annotation, we determined the correlation of the emission parameters of global patterns with the expression of nearby genes annotated to the pattern. Specifically, for each gene considered from the external RNA-seq data[15], we identified the TSS and divided the genomic region surrounding the TSS into 100 bp windows up to 500 kb upstream and downstream of the TSS. This resulted in considering 10,000 equidistant bins between −500 kb and 500 kb from the gene's TSS. We then used the LCL genome annotation of global patterns to determine the global pattern assigned to each of these bins. For each bin, we calculated the Pearson correlation between the gene expression and the emission parameters of the bin's assigned global pattern. This was performed separately for the emission parameters of the three histone modifications. We repeated this process for each gene and averaged the Pearson correlations from each bin representing the distance to a gene's TSS and the assigned global pattern.

For the association of global patterns with co-expression modules, we generated co-expression modules from the corrected LCL RNA-seq data using WGCNA[62]. Following WGCNA pre-processing steps, 219 genes were removed due to missing values and one individual was identified as an outlier during clustering and also removed, resulting in 14,450 genes and 53 individuals used for module calculation. We then ran WGCNA which learned a co-expression network consisting of 19 modules. We tested for association between the module eigengene values and the emission parameters corresponding to each mark. The association was performed separately for each mark. We used an FDR threshold of 5% within each mark.

For the LCL data set, we obtained log2 protein quantification data for 60 of the 75 individuals for a subset of 4371 genes with both gene expression and protein quantification data[38] (Supplementary Data 1). Using the same association framework as the association between global patterns and gene expression, we associated the log2 protein quantification with the global patterns for each mark separately. We used an FDR threshold of 5% within each mark.

### Motif enrichments

We computed enrichment of 602 TF motifs in the ENCODE motifs database using a tool published with the database[48]. Briefly, this tool computes the enrichment of each motif compared to a background created using shuffled control motifs with a confidence interval correction for small counts. Although this database includes both known and computationally discovered motifs, only known motifs, which are more specific to individual TFs, were used. We identified motifs with FDR < 5% and a log2 fold enrichment of at least 1.5.

### Association of global patterns with ASD status

We associated each global pattern in the 90-state ASD model with the phenotype status using a two-sided Mann–Whitney $U$ test. We performed a permutation test to correct for multiple testing, where for each permutation, the samples labels were shuffled and the minimum $p$ value was retained for the null distribution. The adjusted $p$ values were computed as the fraction of $p$ values from the null distribution that were more significant than the observed $p$ values. We used an adjusted $p$ value threshold of 0.05 to identify global patterns significantly associated with ASD status.

We additionally performed logistic regression between the ASD global patterns and the phenotype status. For each global pattern of interest, we performed logistic regression using the emission parameters of the global pattern and the covariates (age, sex, percentage of neuronal cells, brain bank, number of peaks, FRIP, read duplicate fraction, and read alignment fraction). These are the same covariates used to correct the histone signal prior to model training. We performed the logistic regression using statsmodels version 0.13.4 Logit function.

### Statistics and Reproducibility

All statistical tests were performed using the data described in previous sections. In brief, there were 75 LCL histone modification samples, 76 ASD histone modification samples, 54 LCL RNA-seq samples, 51 ASD RNA-seq samples, and 60 LCL protein quantification samples. All association tests (i.e., gQTL, eQTL, association between emission parameters and gene expression, association between emission parameters and protein quantification data) were performed using the MatrixEqtl software v2.2. Significant hits were defined by a Bonferonni corrected $p$ value threshold. The significance of gQTL replication was quantified by comparing the $p$ values for the association between emission parameters and common genetic variants in the BLUEPRINT data set with a uniform distribution using the two-sided Mann–Whitney $U$ Test computed using SciPy v1.9.3. The association between global pattern emission parameters and ASD status was also computed using the two-sided Mann–Whitney $U$ test using SciPy v1.9.3. We additionally tested the association between each global pattern of interest and ASD status with logistic regression accounting for covariates using statsmodels v0.13.4.

## Reporting summary

Further information on research design is available in the Nature Portfolio Reporting Summary linked to this article.

## Data availability

Our work integrates data from a number of published studies, including LCL ChIP-seq and RNA-seq data (http://mitra.stanford.edu/kundaje/portal/chromovar3d/, a link to a copy of the data we used is available through https://github.com/ernstlab/EpiVarIndividuals/)[15], LCL protein data[38], ASD ChIP-seq data (https://www.synapse.org/#!Synapse:syn4587616)[4], and ASD RNA-seq data (https://www.synapse.org/#!Synapse:syn11242290)[34]. A list of samples in each of the LCL data sets, model parameters learned in the paper from the LCL, ASD, and BLUEPRINT data, and model validation results can be found in Supplementary Data 1. The significantly enriched gQTLs in the LCL and ASD data, GREAT enrichments for the gQTLs, significantly enriched eQTLs that overlap with gQTLs in the LCL data set, and BLUEPRINT gQTL replication statistics can be found in Supplementary Data 2. The global pattern annotation of the LCL and ASD models can be found in Supplementary Data 3. The LCL and ASD TF motif enrichments, enrichments with external chromatin states and annotations, and potential trans-regulators can be found in Supplementary Data 4. The significant associations between the LCL and ASD global patterns and gene expression data, co-expression module information and significant associations with global patterns, the significant associations between LCL global patterns and protein quantification data, the overlap between significantly associated genes and proteins, the correlation between gene expression and global patterns, and the association between ASD emission parameters and diagnosis status can be found in Supplementary Data 5.

## Code availability

Code to perform covariate correction, code and data required to run a toy example of the training pipeline, and a README documenting the steps of the training pipeline can be found at https://github.com/ernstlab/EpiVarIndividuals/ and https://zenodo.org/records/14941933[63]. Chromatin state learning was performed using ChromHMM, which is available at https://ernstlab.github.io/ChromHMM/.

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

## Acknowledgements

We would like to acknowledge Daniel Geschwind and Gokul Ramaswami for helping us access the ASD data and for providing feedback on the results. This work was supported in part by the United States National Institutes of Health (DP1DA044371, R01MH109912, U01MH105578, UH3NS104095, and U01HG012079) (J.E.), a Rose Hills Innovator Award (J.E.), and the UCLA Jonsson Comprehensive Cancer Center and Eli and Edythe Broad Center of Regenerative Medicine and Stem Cell Research Ablon Scholars Program (J.E.). J.Z. received supported from a National Science Foundation Graduate Research Fellowship under Grant DGE-1650604 and National Institutes of Health Award T32MH073526. E.M. received support from the NIH Training Grant in Genomic Analysis and Interpretation Award T32HG002536. This study makes use of data generated by the Blueprint Consortium. Funding for the project was provided by the European Union's Seventh Framework Programme (FP7/2007-2013) under grant agreement no 282510 – BLUEPRINT.

## Author contributions

J.Z. and J.E. developed the global pattern framework. J.Z. and E.M. performed analyses. J.E. supervised the project. J.Z., E.M., and J.E. wrote the manuscript. All authors read and approved the final manuscript.

## Competing interests

The authors declare no competing interests.
