## [Transparent peer review file · Communications Biology]

Genome-wide identification and analysis of recurring patterns of epigenetic variation across individuals

Corresponding Author: Dr Jason Ernst

This manuscript has previously been submitted to another journal. This document only contains information relating to versions considered at Communications Biology.

Version 0:

Reviewer comments:

Reviewer #1

(Remarks to the Author)

The authors used a stacked chromatin state model to learn global patterns of epigenetic variation (histone marks here) across individuals to annotate the human genome. The annotated genome was then used for analyzing between across individuals in lymphoblastoid cell lines and across autism spectrum disorder cases and controls in prefrontal cortex tissue, to build a framework to predict transregulators, trans-QTL.

My main concerns are the proposed method was not clearly explained for readers to link with the analysis and results. This makes the fancy model remain a semi-black box that only few people might be interested enough to dig in, while this manuscript is meant for biologists / genomic people to all follow. If we are to assess the novelty of this work, it is not clear to me how much it is different from its based model chromHMM, and there is no clear comparison. In the assessment of significance, the 2 testing data seem to be served only for demonstrating and validating the method. The major findings are not clearly stated/presented.

Nevertheless, I think if all these comments or confusion are resolved and addressed, this can still be an interesting work to attract more general audiences.

1. the definition of "trans-regulators" should be made clearer, as it is the center of this manuscript. In the case of enhancers, do trans-regulators only refer to those effects from the elements/enhancers located at other chromosomes or they are anything distal? Most enhancers identified are from the same chromosomes of the target genes, though.
2. Add literature support for the statement of "a transcription factor may have differential activity across individuals". How strong this variation could be? Observed across species and in plants too?
3. the proposed "stacked" chromatin state model was based on the chromHMM published much earlier. The difference between the two, (or say, the novelty of the stacked model) should be clearly explained, in particular the models/algorithms and their applications to the proposed research.
4. chromHMM should also be clearly explained to general readers, as it is very much the frame of the proposed model.
5. How was the ChIP-seq data processed and used? Were read abundances considered or they are just peak-called as binary values.
6. What happens when some marks are missing?
7. How were all these chip-set data normalized and processed? Were there any filtered or Q.C.ed to ensure the low-quality data are excluded?

8. As the global pattern is the main data format that was generated and analyzed, please explain "global pattern" more clearly? Were they array of 200bp bins and each are the normalized RPKM or 1/0 values? Are there any locations (bins) removed or discarded or specially selected for any reasons?
9. Why are the 3 marks H3K27ac, H3K4me1, H3K4me3 selected? Are they the only 3 available for the LCL data?
10. The conclusion/summary of each analysis should be stated clearly. It reads like a to-do list but the significance or major finding should be stated and highlighted.
11. Figure 1. What are the circle, triangle and square? What are S1, S2 and S3? These should be described in the legend no? Would be helpful if the figure can guide the readers on how this Multivariate HMM converts the raw data on the left to the Emission probabilities on the right (the s is the core step no?) The current figure legend does not really explain why readers see in the figure. Also, if this is a better schematic plot - the subsequent analyses, including motif-TF, GREAT, associating with gene expression, GO should all be included.
12. How was the training and testing done for the model construction?
13. Please explain the "states" of the model specified in this work. Do they refer to combinations of histone mark patterning at specific loci or what? Would that very much depend on the number of marks used? Would the number of 100 states be too many? How was the best number of states decided?
14. Are there any analyses on the real data showing the global patterns are more similar between replicates than between samples? This is critical to ensure what we see is not noise.
15. The number of gQTLs was maximized with the 85-state model -- what does this mean? The authors should keep the technical parts in Methods / Suppl and try to explain their no matter how brilliant ideas to general audiences.
16. Explain how the previous LCL prediction was made and why do we want to specifically compare to that previous prediction? What is the point there?
17. Figure 2C where is the gene expression data on top of the 24 TF enrichment analysis?
18. Were the LCL and ASD data simply used as a demonstration for the pipeline? Why not really test the model on a new disease or phenotypes?
19. The code is poorly presented. No tutorial, no toy data. I suspect anyone will be interested enough to try. As this is more like a method/tool paper, the implementation with the tutorial should be provided in a way most people can test and run.
20. Perhaps also discuss the possibility to incorporate more epigenomic data (DNAm, smRNA, chromatin accessibility).

Reviewer #2

(Remarks to the Author)

Zou and Ernst presented a very interesting new approach to map the genetics of global epigenetic variations and generated some nice results on LCL, brain, immune-related diseases, and autism. The stacked chromatin state model is a novel way to capture the global patterns and reduce data complexity. They identified hundreds of significant gQTL in small samples of less than 80 subjects. This is very encouraging. But there are several major issues to be addressed or clarified:

1. It is not clear why the authors used YRI LCL instead of European LCL. If using European LCL, the results will be more compatible with other downstream analyses like when comparing with the GTEx data. Meanwhile, the paper did not describe whether they controlled population structure in their gQTL mapping. I did not see anything mentioned regarding covariates in the gQTL mapping. It is one of the major concerns.
 2. In the study, the LCL data has 3 histone marks, and the brain data has only one histone mark. I am very curious how the 3-mark models compare to 1-mark models; any advantage of building 3-mark or even more mark models. Since the LCL data has 3 mark. This question can be studied. It will be informative when we consider what to include for building future models. Should try to including more marks? Even more than just histone marks?
 3. The authors compared the chromatin states with gene expression. It will be more interesting to correlate that with coexpression modules, which is another organized expression.
 4. It is intriguing to see that the significantly enriched motifs for the TFs are concentrated in only a few states. There could be different TFs responsible for different states. Why most of the states do not have associated TF motifs? Any thoughts?
 5. The trans-eQTL replication is done by comparing the p values of the eQTL from LCL and from GTEx. Why not compare effect size, which is less sensitive to sample size?
 6. The study did only one replicate assessment for trans-eQTL. Since this study is about the newly-defined chromatin states and related gQTL, it makes more sense to have replicates of these two types of findings.
- Overall, it is a very exciting new procedure. But some critical tech details like covariate controls will be important to have. More data, comparisons, and assessments are needed to prove the robustness findings, which is the essential proof of the validity and value of such a new approach. It is a very promising work.

Reviewer #3

(Remarks to the Author)

In this study, Zou and Ernst proposed an framework for identifying and analyzing global patterns in the genome. The global patterns were generated by a stacked chromatin state model by utilizing histone modification data from different samples. The authors applied this framework to two datasets (lymphoblastoid cell lines and autism spectrum disorder, LCL and ASD). They investigated the associations between SNPs, gene expression, protein abundance data, diagnosis status and global patterns. The analysis covers several aspects and is inspiring for other researchers in the epigenomics field. However, I have some concerns about this study.

1) The global patterns framework is constructed based on the stacked chromatin state model and the output global patterns. Vu and Ernst (Genome Biology, 2022) had proposed a "full-stack ChromHMM model", which provides chromatin state annotation by using over 1,000 datasets from more than 100 cell types. Is there any difference between the "stacked chromatin state model" and "global patterns" in this study and "full-stack ChromHMM model" and "universal chromatin states" in Vu and Ernst?

2) The authors analyzed associations between genetic variation, gene expression, protein abundance, diagnosis status and global patterns. Most of these results were numbers of significant QTLs, genes, motifs, and TFs. More detailed results should be provided to convince the audiences.

2.1) Tables or lists of these statistically significant QTLs, genes, motifs, and TFs should be provided.

2.2) Some examples of these QTLs and genes, especially functionally related to LCL and ASD, should be showed in the manuscript.

2.3) The authors identified two TFs that had gene expression patterns associated with the same global pattern for which their motifs were enriched. Examples of expression and histone modification status for specific target genes regulated by these two TFs are needed.

2.4) In ASD data, the authors showed that two global states were different in ASD cases and controls. Example regions in the genome should be given.

3) The authors integrated multiple sources of public data, including ChIP-seq, RNA-seq, and protein data in LCL and ChIP-seq and RNA-seq data in ASD. This is a good effort, but unfortunately the authors provided three website links for these datasets rather than detailed information.

3.1) The source link for ChIP-seq and RNA-seq data in LCL is a web server. The data download link on this webserver is currently failing. An updated link and description of the data source should be given.

3.2) The authors analyzed hundreds of high-throughput sequencing datasets. Given that only a subset of the individuals in the LCL data set had corresponding protein data, more information about these data sets should be provided.

3.3) What kind of data file (BAM or fastQ format) have the authors obtained from the public high-throughput sequencing data set? If the authors analyzed fastq format files, the mapping software and parameters should be described in the Methods part.

4) The authors mentioned that "there is greater overlap between these two sets of genes than expected by chance." A simple p-value is not enough. A Venn diagram with specific numbers of each set is needed.

5) The descriptions of some metrics are lack of clarity.

5.1) The authors used median Spearman correlation to evaluate the robustness of the chromatin state learned from ChromHMM. The description is too simple to confuse me about how it is calculated. A more detailed version of the Methods description is required, especially when a mathematical formula is used.

5.2) The descriptions of the "average correlation of histone modification LCL emission parameters and gene expression" in Figure S5-7 are quite confusing. A more detailed description about how to calculate these scores is need.

Version 1:

Reviewer comments:

Reviewer #1

(Remarks to the Author)

Zou et al has addressed most of the comments. I have remaining minor questions below,

1. The authors wrote "For the ASD data set, we accounted for age, sex, percentage of neuronal cells, brain bank, number of peaks, fraction of reads in peak (FRIP), read duplicate fraction, and read alignment fraction [4]." How were "the percentage of neuronal cells" derived and what is the range? Computational deconvolution or from experimental assay?

2. Follow (1), since the cell type composition affects sequencing data significantly, I was wondering what is the trait-trait correlation and whether the traits correlate with the case-control prediction in Figure 4.

3. Figure 4 shows the ASD diagnosis with the emission probabilities from the model. Is it possible to have a classifier ROC with this?

Reviewer #2

(Remarks to the Author)

The revision did a great job addressing my previous concerns. Reading through the responses to the other reviewers' comments also made me understand the paper much better. It is a dramatic improvement.

Additional questions I have are:

1. In the association test of co-expression modules with emission probabilities, Figure S11, distinct global patterns have been associated with the same module (like ME9) in similar strength. Only a few modules showed a strong association with multiple patterns, while many other modules did not show any association. What does this mean? It is surprising to me. I thought that different global patterns should link to different co-expression modules in a more dramatic way. The figure shows that their (global patterns') co-expression patterns are more similar than different.

2. I am also surprised to see that a few individual genes, like TP73, correlated with global pattern 14. A global pattern should be more related to global regulation instead of local, regional regulation, which is related to one target gene. Certainly the correlation is not that strong. Therefore, its meaning may not be that important. The co-expression module might be more relevant and, therefore, deserves more space in the main text.

3. Do gQTLs match any eQTLs or other QTLs in LCLs? GREAT analysis did not answer this question.

4. The paper used 200bp non-overlapping bin to run ChromHMM. Is there any concern regarding the effects of the random breakdown of a continuous sequence? Will the results change with the boundaries of bins? Will sliding window give a different result?

Maybe I still have missed something. Hope the authors can help to clarify.

Reviewer #3

(Remarks to the Author)

My comments have been addressed.

Version 2:

Reviewer comments:

Reviewer #1

(Remarks to the Author)

I appreciate the authors for the efforts. The manuscript has improved significantly, and I have no further questions.

Reviewer #2

(Remarks to the Author)

All my concerns are addressed.

Reviewers' comments:

Reviewer #1 (Remarks to the Author):

The authors used a stacked chromatin state model to learn global patterns of epigenetic variation (histone marks here) across individuals to annotate the human genome. The annotated genome was then used for analyzing between across individuals in lymphoblastoid cell lines and across autism spectrum disorder cases and controls in prefrontal cortex tissue, to build a framework to predict transregulators, trans-QTL.

My main concerns are the proposed method was not clearly explained for readers to link with the analysis and results. This makes the fancy model remain a semi-black box that only few people might be interested enough to dig in, while this manuscript is meant for biologists / genomic people to all follow. If we are to assess the novelty of this work, it is not clear to me how much it is different from its based model chromHMM, and there is no clear comparison. In the assessment of significance, the 2 testing data seem to be served only for demonstrating and validating the method. The major findings are not clearly stated/presented.

Nevertheless, I think if all these comments or confusion are resolved and addressed, this can still be an interesting work to attract more general audiences.

We thank the reviewer for the summary and the constructive feedback. We have revised the manuscript to more clearly explain the methodological approach, elaborate on the differences from previous applications of ChromHMM, and highlight its advantages. We also state more clearly the biological findings. We believe our revisions in response to the authors comments have led to a substantially improved manuscript.

1. the definition of "trans-regulators" should be made clearer, as it is the center of this manuscript. In the case of enhancers, do trans-regulators only refer to those effects from the elements/enhancers located at other chromosomes or they are anything distal? Most enhancers identified are from the same chromosomes of the target genes, though.

We thank the reviewer for raising this point and we apologize that this was not clear. By trans-regulators we are referring to transcription factors that bind in locations across the genome and influence multiple genes. The transcription factors can regulate genes either locally or at a distance within the same chromosome or across chromosomes. We have made it more explicit in the introduction that we consider both local and distal target genes. Specifically in the revised manuscript we now state:

“Such transcription factors could potentially act as ‘trans-regulators’, that is proteins affecting the expression of multiple genes in the genome locally or distally. Distal impacts can be on the same or across chromosomes from where the gene is encoded.”

2. Add literature support for the statement of "a transcription factor may have differential activity across individuals". How strong this variation could be? Observed across species and in plants too?

We thank the reviewer for the suggestion. We added citations for studies that describe or harness the variation within a transcription factor across individuals. These citations include Li et al. (Li and Bussemaker 2023), which uses a generalized linear model to predict individual-specific TF activity in an effort to run association tests between TF activity and genotypes suggesting the importance of differential TF activity for phenotypic analyses. This linear model also specifically models the TF regulatory strength using the results of a perturbation analysis as the dependent variable, thus investigating the magnitude of the variation in TF activity. It also includes Deplancke et al. (Deplancke et al. 2016) which reviews how human inter-individual variation impacts TF-DNA interactions, specifically emphasizing the importance of cis-regulatory variation and discussing the impact of trans-effects on binding variability. We also now cite Claussnitzer et al. (Claussnitzer et al. 2014) which shows that both differential and conserved transcription factor binding activity across species assists in the analysis of phenotypic variation, showing that variable TF activity is observed in other species besides human. We also cite Flynn et al. (Flynn et al. 2022) which identifies TF regulators by using the inter-tissue and individual variability of TFs. We did not provide a citation on plant transcription factors as plant analysis does not motivate our study.

In our revised manuscript we now state:

“One biological reason we could expect to observe recurring epigenetic patterns across individuals is that a transcription factor may have differential activity across individuals [24-27].

3. the proposed "stacked" chromatin state model was based on the chromHMM published much earlier. The difference between the two, (or say, the novelty of the stacked model) should be clearly explained, in particular the models/algorithms and their applications to the proposed research.

We thank the reviewer for the suggestion. We have more clearly explained the difference between the stacked modeling and a more traditional concatenated approach of applying ChromHMM. The states of the stacked modeling directly capture patterns of epigenetic variation across individuals as part of the model and annotate the genome based on them. In contrast the concatenated approach when used in this context provides a chromatin state annotation per individual and does not directly capture patterns of epigenetic variation across individuals.

In our revised manuscript we now state in the introduction:

“Another approach that allows for joint analysis of multiple marks is to learn combinatorial and spatial patterns of epigenetic marks (‘chromatin states’) [20] using approaches such as ChromHMM [21], which are then associated with different candidate biological interpretations. In the presence of data from multiple cell types, a common approach is to apply ChromHMM using the concatenated approach [22] where through a virtual concatenation of multiple cell types a

common set of state definitions are learned and used to annotate each cell type. This approach has been previously adapted to histone modification data across multiple individuals by virtually concatenating data across individuals for each data type and learning a chromatin state model using the ChromHMM software [3, 21]. Under this approach, individual-specific genome chromatin state annotations were generated that contain chromatin state assignments for each individual genome-wide. These annotations were then used to manually identify regions with variable chromatin states across individuals [3]. While informative, one understudied aspect of these previous approaches is the relationships between the variable regions, in particular recurring patterns of epigenetic variation across individuals observed in many regions of the genome.

The concatenated approach of the ChromHMM framework, while modeling combinatorial patterns across individuals, does not model any recurring patterns across the genome as each individual is given a unique genome-wide chromatin state annotation. A potential alternative to this concatenated approach is applying the stacked modeling approach of ChromHMM [21]. Under this modeling approach, instead of virtually concatenating the data from multiple individuals, the histone data from multiple individuals is stacked and viewed as separate inputs. This results in a single model based on patterns across all individuals and marks and a single genome-wide annotation shared across all individuals. This stacked approach has been previously applied to data from multiple marks across multiple cell types [23], but has not been applied in the context of multiple individuals in a single cell type.”

4. chromHMM should also be clearly explained to general readers, as it is very much the frame of the proposed model.

We thank the reviewer for this feedback. See our response to the previous comment on how we addressed this.

5. How was the ChIP-seq data processed and used? Were read abundances considered or they are just peak-called as binary values.

We thank the reviewer for the questions. In this paper, we use BAM files produced from prior work. In the methods of our previous and current version, we cite the LCL and ASD data sources, summarize each study’s data processing techniques, and state that we use each study’s processed BAM files. We have adjusted the text to further clarify the data processing, specifically we now state:

“In the LCL data, histone modification signal was mapped for three marks (H3K27ac, H3K4me1, H3K4me3) in 75 individuals using procedures described in [15]. Briefly, reads were aligned to personal genomes and filtered for unmapped reads, duplicates, unpaired reads, and reads with low quality. The BAM files from this work were used for this paper. We did not perform any additional quality control beyond what was previously applied. In the ASD data, H3K27ac signal was mapped in the prefrontal cortex tissue for 93 individuals. Reads were mapped to the human reference genome and duplicate reads were filtered. Extensive sample quality control was

performed in [4], resulting in a number of samples excluded. We used BAM files for the same 76 individuals that were used in previous work after quality control [4]. For this data, we also did not perform any additional quality control beyond what was previously applied. Only autosomes were included in this analysis and we used the hg19 genome assembly.”

Furthermore, we consider read abundances in ChromHMM training. As we explain in the methods, the ChromHMM input splits the entire genome into equal sized bins, with each entry representing the number of aligned reads that fall within that bin. These read counts were then adjusted using a quasi-poisson regression model and quantile normalization. Thus, the final input used to train the HMM model was adjusted read abundances.

The relevant methods text we had on this in our previous and current version is:

“We quantified the number of reads falling within each 200bp non-overlapping bin in the genome using the BinarizeBam command in the ChromHMM software package (version 1.11) [21]. For each bin, we then fit a quasi-poisson regression model, where the dependent variable was the number of reads in the bin and the independent variables were standardized known covariates [55]. Quasi-poisson regression handles both over and under-dispersion of count data and has been used in related applications [56]. We quantile normalized the read counts so that the distribution of counts across bins in the genome was the same for each individual. For each data set, we regressed out the effect of the known covariates that were used in previous work. For the LCL data set, we accounted for sex, genotyping method, number of reads and relative strand correlation (RSC) [15]. We performed the correction separately for each mark in the LCL dataset. For the ASD data set, we accounted for age, sex, percentage of neuronal cells, brain bank, number of peaks, fraction of reads in peak (FRIP), read duplicate fraction, and read alignment fraction [4]. We then binarized the corrected histone modification count data according to the procedures used in ChromHMM using the BinarizeSignal command with default parameters [21]. We used binarized read count data as input to learn the parameters to a stacked ChromHMM model.”

6. What happens when some marks are missing?

We thank the reviewer for the question. If a specific histone modification is missing entirely then any unique information offered by that specific mark will not be captured. If multiple marks are considered and not every mark is mapped in each individual, the stacked chromHMM model can still be directly applied on this incomplete histone modification data. This is a strength of the framework compared to the concatenated approach and any method that defines investigates individual specific patterns.

We added a statement in the discussion indicating this strength of the stacked model:

“Another potential advantage of the stacked modeling is that in general they should be less sensitive to missing epigenetic marks in an individual compared to approaches that define individual specific chromatin state annotations.”

Moreover, we have added a supplementary analysis investigating the set of patterns learned based on subsets of the three histone modifications. This analysis allows us to quantify the differences in the learned global patterns when histone modifications are missing. Specifically we use the EvalSubset command in the ChromHMM software to evaluate what proportion of the original genome annotation based on the 85-state model can be recovered when using the model training all marks, but only using subsets of the available marks for the decoding. We found that using a single mark only partially recovers the original genome annotation based on the full set of marks, especially for enhancer and promoter-like states. Using two marks for any pair of marks increases the proportion of recovery, particularly in active TSS, promoter, and enhancer regions of the genome, but still does not allow for full recovery of the annotation based on the full set of marks. This shows that while using all three marks during training captured additional information, informative global patterns are still learned when some marks are missing.

Relevant text added to the Results: “We additionally investigated to what extent the data from subsets of the three histone modifications are sufficient to recover the original genome annotation from the 85-state model trained on all three marks (Methods).. We found that annotations based on decoding with a single mark had only moderate ability to recover the original genome annotation based on the full set of marks, especially for enhancer and promoter-like patterns. Regions of the genome where the original annotation was not recovered were primarily annotated into global patterns enriched for heterochromatin and transcribed associated chromatin states (Figure S6). Using any two marks increased the proportion of recovery, particularly in active TSS, promoter, and enhancer-like patterns, but still did not allow for full recovery of the annotation based on using all three marks (Figure S7).

Overall, the learned LCL global patterns were highly enriched for promoter and enhancer-like patterns which agreed with external annotations and were informed by patterns of variation across the three LCL histone modifications.”

Relevant text added to the Methods: **“LCL annotation recovery with subsets of histone modifications**

We evaluated the annotation recovery with subsets of marks from the LCL model using the EvalSubset command in the ChromHMM software (Version 1.11) [21]. This command evaluates the proportion of the annotations based on the full input that can be recovered when using a subset of the input for decoding. The output consists of a confusion matrix indicating the proportion of the genome annotated into each global pattern based on the full and subset of marks. The diagonal of the confusion matrix represents the proportion of the genome annotation assigned to the same pattern when using a subset of marks relative to when using all three marks for decoding.”

Supplementary figures and text from figure legends:

“Recovery of global pattern genome annotation using one histone modification. A-C) Heatmaps displaying the confusion matrices when using only data from a single mark: A) H3K4me1, B) H3K4me3, and C) H3K27ac to annotate global patterns based on the model learned using all three marks. Each row corresponds to a state in the original model trained on all three histone modifications. Each column represents the percentage of the genome annotation assigned to each state when using the indicated mark for decoding. The diagonal of the confusion matrix represents the percentage of the original annotation that was recovered based on the single mark. The color bar labels each state by the chromatin state annotation

similar to Figure 2A. D-F) Boxplots showing the diagonal values of the confusion matrices when using only data from a single mark on D) H3K4me1, E) H3K4me3, and F) H3K27ac to annotate global patterns with the model learned using all three marks. Each box corresponds to the global patterns enriched for a chromatin state annotation. The boxes use the same color scheme as the color bars in A-C and Figure 2A.”

“Recovery of global pattern genome annotation using two histone modifications. A-C) Heatmaps displaying the confusion matrices when using only data from pairs of marks: A)

H3K27ac and H3K4me1, B) H3K27ac and H3K4me3, and C) H3K4me1 and H3K4me3 to annotate global patterns based on the model learned using all three marks. Each row corresponds to a state in the original model trained on all three histone modifications. Each column represents the percentage of the genome annotation assigned to each state when using the indicated mark for decoding. The diagonal of the confusion matrix represents the percentage of the original annotation that was recovered based on the pair of marks. The color bar labels each state by the chromatin state annotation similar to Figure 2A. D-F) Boxplots showing the diagonal values of the confusion matrices when using only data from D) H3K27ac and H3K4me1, E) H3K27ac and H3K4me3, and F) H3K4me1 and H3K4me3 to annotate global patterns with the model learned using all pairs of marks. Each box corresponds to the global patterns enriched for a chromatin state annotation. The boxes use the same color scheme as the color bars in A-C and Figure 2A.”

7. How were all these chip-set data normalized and processed? Were there any filtered or Q.C.ed to ensure the low-quality data are excluded?

We thank the reviewer for the questions. We use covariate-corrected read counts as input to the ChromHMM software. For each bin, we then fit a quasi-poisson regression model, where the dependent variable was the number of reads in the bin and the independent variables were standardized known covariates. We regressed out the effect of the known covariates for each data set that were used in previous work. For the LCL data set, we accounted for sex, genotyping method, number of reads and relative strand correlation (RSC). For the ASD data set, we accounted for age, sex, percentage of neuronal cells, brain bank, number of peaks, fraction of reads in peak (FRIP), read duplicate fraction, and read alignment fraction. This information is described in more detail in the “Learning epigenetic patterns across individuals using a stacked ChromHMM model” Methods section.

For both the LCL and ASD data, the final samples in the published work were extensively QC-ed by filtering for unmapped reads, duplicates, unpaired reads, and reads with low quality. Thus, we did not perform any additional sample QC. We have made this clearer in the Methods. The relevant text describing the LCL data in the methods now states:

“Briefly, reads were aligned to personal genomes and filtered for unmapped reads, duplicates, unpaired reads, and reads with low quality. The BAM files from this work were used for this paper. We did not perform any additional quality control beyond their methods.”

The relevant text describing the ASD data now states:

“We used BAM files for the same 76 individuals that were used in previous work after quality control [4]. We did not perform any additional quality control beyond their methods.”

8. As the global pattern is the main data format that was generated and analyzed, please explain "global pattern" more clearly? Were they array of 200bp bins and each are the normalized RPKM or 1/0 values? Are there any locations (bins) removed or discarded or specially selected for any reasons?

We thank the reviewer for the questions and apologize that this was not clearer in the original version.

There are two main data formats that were generated and analyzed in our paper.

1. Global patterns are the hidden states learned using the stacked chromatin state model. They can be represented using the emission probabilities, which represent the probability of observing the mark in a specific individual given the hidden state.
2. The genome segmentation and annotation contains global pattern assignments for 200bp non-overlapping bins in the genome. No bins are removed or discarded. The global patterns were learned based on binarized data. The binarized data was determined by taking into account the sequencing depth of each individual.

We have made this clearer when introducing the global patterns of epigenetic variation:

“The hidden states of the stacked model are learned on data from multiple individuals in one cell type and represent recurring patterns across individuals and potentially also marks, which we will refer to as ‘global patterns’”

We have additionally made this clearer in the first paragraph of the results section:

“Unlike the standard use of ChromHMM trained on a single individual, in this framework, each hidden state learned corresponds to a combinatorial and spatial pattern across individuals and potential also marks, which we call a ‘global pattern’. Each global pattern has emission probabilities corresponding to the probability of observing a mark in a specific individual given a global pattern, which we use for downstream analysis. After the global patterns are learned, we annotate the genome at 200bp resolution with the most likely hidden state of the HMM. This singular genome-wide annotation is universal to all individuals.”

9. Why are the 3 marks H3K27ac, H3K4me1, H3K4me3 selected? Are they the only 3 available for the LCL data?

We thank the reviewer for the question. We decided to focus on histone modifications since these are the most common input to ChromHMM and their relatively broad and genomewide nature is the type of data ChromHMM was designed to capture. In addition, histone modification data was available for LCL and ASD. Among histone modifications, these three were the only histone modifications available in the data set used (described in the “Histone modification data” Methods section). Other histone modifications available in LCL have only been profiled in a more limited set of individuals. These three marks are associated with promoters and enhancers, so are expected to be informative for capturing epigenetic variation across individuals for regulatory regions. While we used LCL DNase I hypersensitivity data as part of a post analysis step, including it as part of the modeling could be an avenue for future investigation. We modified the discussion section to further elaborate on this potential avenue.

We updated the results section describing the LCL data to further elaborate why these specific histone modifications are interesting to study:

“We used this LCL data to showcase the utility of global patterns in a cell type that has previously been used to study histone modification variation across individuals [5, 15, 39] and data for three different histone modifications that have previously been shown to be associated with different types of regulatory elements were available [12, 15, 40, 41].”

We updated the discussion to make the possible extension beyond histone modification data more clear:

“The framework we have used is general and can be applied or adapted to other datasets from different cell types, phenotypes, and other types of epigenetic data such as on DNA methylation or chromatin accessibility.”

10. The conclusion/summary of each analysis should be stated clearly. It reads like a to-do list but the significance or major finding should be stated and highlighted.

We thank the reviewer for the feedback and suggestions. We have revised the text to better highlight our main points and their significance. We provide a few examples below of these statements.

We explain that the significant association of global patterns with gene expression and protein quantification data further supports the global pattern’s biological relevance.

“Overall, LCL global patterns based on epigenetic variation across individuals were associated with large-scale molecular variation across individuals as determined by other assays not used during model training, further supporting their biological relevance.”

We explain that the strong enrichment in promoter and enhancer regions supports the utility of global patterns.

“The majority of global patterns were most highly enriched for enhancer and promoter chromatin states. We define these global patterns as enhancer- and promoter-like global patterns, respectively. The high correspondence with enhancer and promoter chromatin states was expected given the histone modifications used to learn the model are known to be associated with enhancer and promoter activity (Figure 2A) and supports the relevance of the annotations for analysis of epigenetic variation at regulatory regions.”

We explain that the significantly enriched TF motifs are primarily in active regions of the genome where motif activity would be expected.

“These TF motifs are primarily enriched in global patterns corresponding to promoter and active TSS chromatin state annotations, with some corresponding to enhancer states (Figure 2A, Supplementary Data 5). This motif enrichment in putative regulatory regions of the genome provides orthogonal support of the likely regulatory nature of these genomic regions.”

We explain why the TF motif enrichment analysis is informative

“Overall, TF motif enrichment analysis for global patterns and in some cases combined with gene expression associations can assist in the identification of potential trans-regulators.”

11. Figure 1. What are the circle, triangle and square? What are S1, S2 and S3? These should be described in the legend no? Would be helpful if the figure can guide the readers on how this Multivariate HMM converts the raw data on the left to the Emission probabilities on the right (the s is the core step no?) The current figure legend does not really explain why readers see in the figure. Also, if this is a better schematic plot - the subsequent analyses, including motif-TF, GREAT, associating with gene expression, GO should all be included.

We thank the reviewer for the suggestions.

We have added labels for the individuals and marks to Figure 1. The circle, triangle, and square correspond to unique histone modifications and P1, P2, and P3 correspond to global patterns as we now indicate in the figure.

We have also added some examples of downstream analyses to the figure, including the gQTL analysis, correlation of emissions with gene expression and protein expression, association with disease status, and enrichments for TF motifs and other external annotations. We excluded the GREAT analysis, since this was primarily included as an orthogonal validation of the gQTLs. We indicate in the figure legend that these are potential uses of the emission parameters and genome annotation of global patterns and that our framework is flexible to other analyses.

We believe that illustrating the process of going from raw data to emission probabilities would distract the main focus and novelty of this paper and overcomplicate it so our preference is not to include those details in the figure.

“**Method Overview.** We trained a stacked multivariate HMM model using genome-wide histone modification signal from multiple individuals and for one dataset also marks (“Genome-wide histone modification signal”) using ChromHMM. This model learns global patterns of epigenetic variation that recur in many regions of the genome (“Emission Probabilities”). The model learning is agnostic to the mark or individual label of each dataset. The emission probabilities learned correspond to the probability of observing a presence calling for each data set conditioned on being in each hidden state. We used the model learned to annotate the genome according to these patterns at 200bp resolution (“Genome annotation”). We then used both the model emission parameters and global pattern annotations can be used for various downstream analyses, including but not limited to the examples illustrated in the figure. We associated the emission probabilities of the global patterns with other measures of molecular variation or phenotypic status. Additionally we computed for locations annotated to specific global patterns enrichments for external annotations. Motif example was generated with [38].”

12. How was the training and testing done for the model construction?

We thank the reviewer for the question. Our application of Hidden Markov models was unsupervised machine learning, thus as we are not using labeled training data there is no separate held out testing dataset. Instead, we are learning patterns in the data and interpreting what they could mean biologically.

Nevertheless, we do some analyses to ensure that the model we learn is robust. We trained models using different subsets of data and found similar global patterns learned between these models (Supplementary Figure 2). We also performed pairwise correlations of the marks used to train the LCL model (Supplementary Figure 1). High correlations were observed even though the models were agnostic to the mark and individual labels in the training process. This suggests that a global pattern is less likely to reflect technical issues with the ChIP-seq experiments and more likely to be associated with differences at the sample level.

We also compare the global patterns with a number of data sets not used for training including DNA, gene expression, and protein expression data (LCL only) to show that similar patterns across individuals are observed in multiple data sets.

In our updated method overview figure, we illustrate our framework for training ChromHMM and highlight some downstream analyses we did. These changes can be found in our response to comment 11 above.

13. Please explain the "states" of the model specified in this work. Do they refer to combinations of histone mark patterning at specific loci or what? Would that very much depend on the number of marks used? Would the number of 100 states be too many? How was the best number of states decided?

We thank the reviewer for the questions. The "states" of the model represent a combinatorial and spatial pattern of histone modification peaks across individuals and correspond to the global patterns. The states themselves do not represent histone modification combinations at a specific genomic loci, but rather common histone modification patterns across individuals seen in multiple locations throughout the genome. Each state has corresponding probabilities of observing a histone modification in each individual along with transition probabilities that can capture spatial information which are used to annotate the genome into state. This segmentation and annotation process labels the specific loci with the global pattern corresponding to the most likely state at the position.

In the introduction, we modify the text introducing the chromatin states, making clear that they represent individual specific signals: "Under this approach, individual-specific genome chromatin state annotations were generated that contain chromatin state assignments for each individual genome-wide."

"The concatenated approach of the ChromHMM framework, while modeling combinatorial patterns across individuals, does not model any recurring patterns across the genome as each individual is given a unique genome-wide chromatin state annotation."

We also modified the text where we introduce global patterns, making clear that they represent patterns across potentially marks and individuals: "The hidden states of the stacked model are

learned on data from multiple individuals in one cell type and represent recurring patterns across individuals and potentially also marks, which we will refer to as 'global patterns'."

We also described this in the Results section with the following modified text: "Unlike the standard use of ChromHMM trained on a single individual, in this framework, each hidden state learned corresponds to a combinatorial and spatial pattern across individuals and potential also marks, which we call a 'global pattern.' Each global pattern has emission probabilities corresponding to the probability of observing a mark in a specific individual given a global pattern, which we use for downstream analysis. After the global patterns are learned, we annotate the genome at 200bp resolution with the most likely hidden state of the HMM. This singular genome-wide annotation is universal to all individuals."

We additionally changed the title of the paper to not include "chromatin state" terminology to avoid confusion between previous chromatin states and the newly defined global patterns: "Genome-wide identification and analysis of recurring patterns of epigenetic variation across individuals"

The number of states is a hyperparameter selected by the user. Any number of hidden states can be used. The selected number of states impacts how specific a combination of histone modifications across individuals can be represented and the effectiveness of the specific number of hidden states will depend on the desired downstream analysis. Too low a number of states may result in very general patterns that make the annotation and segmentation less informative and too high a number of states may result in overly specific patterns.

We specifically selected the number of states by performing a parameter search from 5 to 100 states in increments of 5. For each trained model, we perform an association test to identify gQTLs (global pattern quantitative loci), which represent a measure of the genetic basis of the epigenetic patterns identified by the learned states. We selected the number of states that maximized the number of significant gQTLs. We describe this process at various points in the paper.

The relevant text from the LCL Results section on this is: "To investigate the genetic basis of global epigenetic patterns across individuals, we performed what we termed global pattern quantitative trait association analysis. For each model that we had learned, we associated common variants with minor allele frequency (MAF) greater than 0.05 in the data set [15 with the emission parameters of each global pattern to identify significantly associated ($p_{adj} < 0.05$) global pattern quantitative trait loci ('gQTLs'). The number of gQTLs was maximized with the 85-state model, (2945 gQTLs, Figure S3), which we selected for further analysis."

The relevant text from the ASD Results section on this is: "We used the same procedures as the LCL data set to preprocess the data, train models, and select the number of hidden states except that we accounted for a larger set of known covariates that were used in previous differential peak analyses (Methods) [4]. We selected the 90-state model (Figure 3) for

follow-up analyses, which maximized the number of gQTLs (Figure S18}, Supplementary Data 5) with 2229 gQTLs.”

The relevant text from Methods on this is: “To select the number of hidden states, which correspond to the number of global patterns, we first trained models using between 5 and 100 hidden states in increments of 5 states. We then associated common variants (MAF>5%) in the human genome with the emission parameters to identify gQTLs and chose the number of hidden states that maximized the number of gQTLs.”

14. Are there any analyses on the real data showing the global patterns are more similar between replicates than between samples? This is critical to ensure what we see is not noise.

We thank the reviewer for the question. Data from replicate ChIP-seq experiments is not available so we did not directly analyze replicate data. However, we have shown there is substantial correlation between emission parameters for pairs of marks within the same state in LCL and also with gene expression for both LCL and ASD data. Observing high correlation on these tasks is more difficult than between technical replicates as some differences are also expected because of the nature of different marks. To clarify this point we revised the result section explaining the correlation between pairs to make our validation methodology more clear.

Relevant text from Results section: “As an initial validation of the models we tested whether the global patterns were internally consistent across pairs of histone modifications. As pairs of these three marks are known to occur in the genome, we expect the emission parameters for pairs of marks to be correlated across individuals in many of the states. As these patterns are from models learned agnostic to mark labels during training, correlation between pairs would support that the states are capturing reproducible cross-individual signal variation.”

15. The number of gQTLs was maximized with the 85-state model -- what does this mean? The authors should keep the technical parts in Methods / Suppl and try to explain their no matter how brilliant ideas to general audiences.

We thank the reviewer for the question and feedback. As described in response to comment 13, we select the number of hidden states based on the maximum number of global pattern quantitative loci (gQTL). These gQTLs are identified via association between common variants and global pattern emission parameters. In essence this is an association between genetic information for each individual and epigenetic variation across the individuals. We use the number of gQTLs as an approximation for the global pattern’s utility in downstream analyses. The technical details of performing the gQTL analysis is described in the Methods. A brief description of the gQTL analysis and reasoning behind the state selection using the gQTLs is described in the Results with the relevant text shown in our response to comment 13.

In order to separate some technical details between the main Results and Methods sections and clarify our motivations for technical choices for both the gQTL analysis and our other

analyses, we removed definitions from the Results and expanded higher-level descriptions in the Results. We show examples of these changes below.

We added further motivation for the gQTL analysis to and removed technical details from the description of the gQTL analysis in the Results section.

Relevant added text from Results: “To maximize the number of discovered gQTLs we focused our analysis on the model with the number of hidden states that maximized this quantity (Methods).”

Relevant deleted text from Results: “We used the gQTLs obtained as the foreground and the whole genome as the background.” This information involves technical details about how the GREAT enrichment analysis was performed and can be found in the Methods.

We moved the description of the validation process involving the median Spearman correlation between pairs of histone modifications.

Relevant modified text in Results with technical details removed: “To investigate this correlation, we performed an analysis to obtain the median Spearman correlation of emission parameters for each pair of histone modifications (Methods).”

Relevant text added to Methods: “For each model trained with between 5 to 100 hidden states, we evaluated the pairwise correlations of emission parameters between marks for the same pattern. For each pair of histone modifications (H3K27ac-H3K4me1, H3K27ac-H3K4me3, H3K4me1-H3K4me3), we performed the following analysis to obtain the median Spearman correlation of emission parameters. For each global pattern learned by the model, we first computed the Spearman correlation between the emission parameters for the first mark in the pair with the emission parameters for the second mark. After computing this correlation for each global pattern, we took the median Spearman correlation across all global patterns. We compared the median Spearman correlations for all models trained using between 5 and 100 hidden states and compared the results.”

16. Explain how the previous LCL prediction was made and why do we want to specifically compare to that previous prediction? What is the point there?

We thank the reviewer for the questions, though we are unsure which previous LCL prediction is being referenced in them.

If the questions reference the two examples of predicted trans-regulators for LCL (TP73 and CREB5), we compare these predicted trans-regulators with their known biological function to support our findings. Both have been validated to have trans-regulatory activity related to immune function (Ren et al. 2020, Wen et al. 2010).

If this comment refers to the previous Kasowski et al. study that uses a “concatenated” ChromHMM model to differences in chromatin states across individuals using LCL data, we expanded our discussion in the introduction of the relationship of the stacked modeling approach used in this paper to the concatenated approach used in that paper. A “concatenated”

ChromHMM model generates individual-specific genome-wide chromatin state annotations, but does not directly identify recurring patterns across the genome. In contrast the “stacked” ChromHMM approach directly identifies recurring patterns of epigenetic variation across individuals and annotates genomic regions associated with them.

Relevant updated text from introduction: “This approach has been previously adapted to histone modification data across multiple individuals by virtually concatenating data across individuals for each data type and learning a chromatin state model using the ChromHMM software [3, 21]. Under this approach, individual-specific genome chromatin state annotations were generated which contain chromatin state assignments for each individual genome-wide. These annotations were then used to manually identify regions with variable chromatin states across individuals [3]. While informative, one understudied aspect of these previous approaches is the relationships between the variable regions, in particular recurring patterns of epigenetic variation across individuals observed in many regions of the genome.

The concatenated ChromHMM framework, while modeling combinatorial patterns across individuals, does not model any recurring patterns across the genome as each individual is given a unique genome-wide chromatin state annotation. A potential alternative to this “concatenated” approach is training a stacked ChromHMM model. Instead of virtually concatenating the data from multiple individuals, the histone data from multiple individuals is “stacked” and viewed as separate inputs. This results in a single model based on patterns across all individuals and marks and a single genome-wide annotation shared across all individuals [22]. This stacked approach has been previously applied to data from multiple marks across multiple cell types [23] and can be applied in the context of this study across multiple individuals in the same cell type.”

17. Figure 2C where is the gene expression data on top of the 24 TF enrichment analysis?

We thank the reviewer for the question. We have clarified that only the 24 TFs whose gene expression levels are significantly associated with the emission parameters of a global pattern and have its motifs enriched in locations annotated to one of the global patterns.

We have added a supplementary figure with the gene expression data from each individual corresponding to the 24 TFs. We showed the association between emissions and gene expression data in the supplementary figures S8-S10.

"Gene expression data for 24 TFs in LCLs across individuals. Gene expression data for the 24 TFs shown in Figure 2C with motifs enriched in at least one global pattern (FDR<5%, $-\log_2(\text{fold enrichment}) > 1.5$) and gene expression associated with global patterns (FDR<5%). Each column represents one of the 54 individuals with available gene expression data corresponding to the order of individuals in Figure 2. Each row corresponds to one of the 24 genes. Four genes had substantially higher gene expression across individuals and are displayed in a separate heatmap with a different color scale for better visualization. Color bars represent normalized TPM levels."

18. Were the LCL and ASD data simply used as a demonstration for the pipeline? Why not really test the model on a new disease or phenotypes?

We thank the reviewer for the questions. Yes, LCL and ASD were selected to demonstrate the analytical approach. We selected the LCL data based on the availability of histone modification data in three marks (H3K4me1, H3K4me3, H3K27ac) across a relatively large number of individuals along with matched gene expression and genotype information which allowed us to study how combinations of these marks are associated with active regions of the genome. We selected the ASD based on the availability of histone modification data in one histone modification (H3K27ac) across a relatively large number of individuals with matched gene expression, genotype, and case/control information which allowed us to study the relationship between learned global patterns and disease. LCL has been used in a number of other studies analyzing epigenetic variation across individuals, while ASD allows us to demonstrate the

approach in the context of a complex phenotype. Assuming the availability or collection of additional data, future work could apply the approach in the context of other diseases or phenotypes. We modified the text to better highlight why we chose these datasets.

Relevant text in LCL Results: “We used this LCL data to showcase the utility of global patterns in a cell type that has previously been used to study histone modification variation across individuals [5, 15, 39] and data for three different histone modifications that have previously been shown to be associated with different types of regulatory elements were available [12, 15, 40, 41].”

Relevant text regarding ASD in Introduction: “Previous studies have identified numerous molecular features, including RNA expression, RNA splicing, and histone modifications, that differ between ASD cases and controls [4, 33-37]. We show that global patterns are also associated with diagnosis status, reflecting the known associations with molecular features.”

Relevant text in ASD Results: “In order to investigate global patterns in the context of a complex disorder, we focused on ASD. Various molecular features including the histone modification H3K27ac have been mapped in ASD case along with control individuals [4].”

19. The code is poorly presented. No tutorial, no toy data. I suspect anyone will be interested enough to try. As this is more like a method/tool paper, the implementation with the tutorial should be provided in a way most people can test and run.

We thank the reviewer for this feedback. We added a toy example to the Github page. The Snakefile and toy data outline the scripts and data formats used to train the stacked ChromHMM model. We have added a readme to the Github page which describes the process, including data and software requirements.

Revised code availability text: “Code, a toy example of the training pipeline, and a README documenting the steps of the training pipeline can be found at <https://github.com/ernstlab/EpiVarIndividuals>.”

20. Perhaps also discuss the possibility to incorporate more epigenomic data (DNAm, smRNA, chromatin accessibility).

We thank the reviewer for the suggestion. We extended our statement in the discussion regarding how the analysis framework is general by giving examples of how it could be extended.

Relevant text: “The framework we have used is general and can be applied or adapted to other datasets from different cell types, phenotypes, and other types of epigenetic data such as DNA methylation and chromatin accessibility. These potential extensions would allow for further analysis and understanding of epigenetic variation across individuals and its relationship with complex phenotypes.”

Reviewer #2 (Remarks to the Author):

Zou and Ernst presented a very interesting new approach to map the genetics of global epigenetic variations and generated some nice results on LCL, brain, immune-related diseases, and autism. The stacked chromatin state model is a novel way to capture the global patterns and reduce data complexity. They identified hundreds of significant gQTL in small samples of less than 80 subjects. This is very encouraging. But there are several major issues to be addressed or clarified:

We thank the reviewer for the summary and the positive comments. We also thank the reviewer for the constructive comments to which we respond to below and led to a substantially improved manuscript.

1. It is not clear why the authors used YRI LCL instead of European LCL. If using European LCL, the results will be more compatible with other downstream analyses like when comparing with the GTEx data. Meanwhile, the paper did not describe whether they controlled population structure in their gQTL mapping. I did not see anything mentioned regarding covariates in the gQTL mapping. It is one of the major concerns.

We thank the reviewer for these comments.

We chose to use the YRI LCL dataset as it had ChIP-seq data for three different histone modifications for a relatively large number of marks. In addition there are other datasets profiled in the same individuals, including genotypes, gene expression, and protein quantification that we could incorporate in downstream analyses.

In the revised manuscript, we now include a gQTL replication analysis using this BLUEPRINT data (see response to comment 6 below). We also note that we have removed the trans-eQTL replication and instead performed the gQTL replication with the BLUEPRINT data. We expand upon our reasoning for removing the trans-eQTL analysis in our response to comment 5. Thus, the motivation for using European LCL data to be more compatible with the GTEx replication is no longer present.

In terms of covariates, we controlled for covariates by regressing out the effects of known covariates prior to the QTL analysis. This is described in the Methods section with the following text: “We regressed out the effect of the known covariates for each data set that were used in previous work. For the LCL data set, we accounted for sex, genotyping method, number of reads and relative strand correlation (RSC) [15]. We performed the correction separately for each mark in the LCL dataset. For the ASD data set, we accounted for age, sex, percentage of neuronal cells, brain bank, number of peaks, fraction of reads in peak (FRIP), read duplicate fraction, and read alignment fraction [4].”.

We previously did not control for population structure in the gQTL analysis due to all the individuals being from the Yoruba population. However, in response to this comment, we have

updated the gQTL analysis to include the first 20 principal components of the genotyping data as covariates. Adding this resulted in a new set of gQTLs. These gQTLs discovered using the genotyping PCs as covariates showed highly significant overlap with the original gQTLs discovered (fold enrichment 1022, binomial one-sided p-value < 1e-300). The Venn diagram below shows the size of the overlaps between the new gQTL set and the version without covariates used in our original submission.

Number and Overlap of gQTLs: LD Pruned

We note the 85-state model still maximizes the number of resulting gQTLs when the genotyping PCs are used as covariates, thus the model selection and subsequent analyses using the model remains unchanged. To account for the changes in the set of gQTLs, we have adjusted the GREAT enrichment analysis and updated the figure showing the number of gQTLs for the models with different numbers of hidden states. While the exact statistics related to the GREAT and gQTL enrichments have changed, the overall conclusions have remained the same. The new set of gQTLs are still significantly enriched for terms related to the lymphoblastoid cell line and immune system function.

Relevant text from the LCL Results section now states:

“The number of gQTLs was maximized with the 85-state model, (2945 gQTLs, Figure S3, which we selected for further analysis. In this 85-state model, 36 states were associated with at least one gQTL.”

“The “regulation of lymphocyte activation” gene ontology (GO) term was significantly enriched for the gQTLs (FDR < 5%, Supplementary Data 1).”

Relevant text now included in the Methods section:

“For the LCL data, we included the first 20 genotyping principal components (PCs) as covariates when performing the associations. The PCs were calculated using the Plink software version v1.90b6.24 [59].”

2. In the study, the LCL data has 3 histone marks, and the brain data has only one histone mark. I am very curious how the 3-mark models compare to 1-mark models; any advantage of building 3-mark or even more mark models. Since the LCL data has 3 mark. This question can be studied. It will be informative when we consider what to include for

building future models. Should try to including more marks? Even more than just histone marks?

We thank the reviewer for the questions. The stacked ChromHMM model can be trained on only one histone modification and reveal informative patterns of epigenetic variation, as demonstrated with the ASD data. However, training on multiple marks provides the opportunity to potentially learn a more informative set of patterns that could capture additional putative regulatory regions or allow making additional distinctions between currently captured regions.

To more systematically investigate this we performed an analysis that evaluated what proportion of the original genome annotation based on the 85-state model can be recovered when using the model trained on all marks, but only using subsets of the available marks for the decoding. For this we used the EvalSubset command in the ChromHMM software. We found that using a single mark only partially recovers the original genome annotation based on the full set of marks, especially for enhancer and promoter-like states. Using two marks for any pair of marks increases the proportion of recovery, particularly in active TSS, promoter, and enhancer regions of the genome, but still does not allow for full recovery of the annotation based on the full set of marks. This shows that including all three marks during training captured additional information. As described below, we now include results from this ablation study in the results section. We also describe the use of the EvalSubset command in the methods.

We also added text to the discussion about the possibility of including other types of epigenetic data into the model.

Relevant text added in discussion: “The framework we have used is general and can be adapted to other datasets from different cell types, diseases, and other types of epigenetic data such as DNA methylation or chromatin accessibility. These potential extensions would allow for further analysis and understanding of epigenetic variation across individuals and its relationship with phenotypes.”

Relevant text added to the Results: “We additionally investigated to what extent the data from subsets of the three histone modifications are sufficient to recover the original genome annotation from the 85-state model trained on all three marks (Methods). We found that annotations based on decoding with a single mark had only moderate ability to recover the original genome annotation based on the full set of marks, especially for enhancer and promoter-like patterns. Regions of the genome where the original annotation was not recovered were primarily annotated into global patterns enriched for heterochromatin and transcribed associated chromatin states (Figure S6). Using any two marks increased the proportion of recovery, particularly in active TSS, promoter, and enhancer-like patterns, but still did not allow for full recovery of the annotation based on using all three marks (Figure S7).

Overall, the learned LCL global patterns were highly enriched for promoter and enhancer-like patterns which agreed with external annotations and were informed by patterns of variation across the three LCL histone modifications.”

Relevant text added to the Methods: **LCL annotation recovery with subset of histone modifications**

We evaluated the annotation recovery with subsets of marks from the LCL model using the EvalSubset command in the ChromHMM software (Version 1.11) [21]. This command evaluates the proportion of the annotations based on the full input that can be recovered when using a subset of the input for decoding. The output consists of a confusion matrix indicating the proportion of the genome annotated into each global pattern based on the full and subset of marks. The diagonal of the confusion matrix represents the proportion of the genome annotation assigned to the same pattern when using a subset of marks relative to when using all three marks for decoding.”

Supplementary figures and text from figure legends:

“Recovery of global pattern genome annotation using one histone modification. A-C) Heatmaps displaying the confusion matrices when using only data from a single mark: A) H3K4me1, B) H3K4me3, and C) H3K27ac to annotate global patterns based on the model learned using all three marks. Each row corresponds to a state in the original model trained on all three histone modifications. Each column represents the percentage of the genome annotation assigned to each state when using the indicated mark for decoding. The diagonal of the confusion matrix represents the percentage of the original annotation that was recovered based on the single mark. The color bar labels each state by the chromatin state annotation

similar to Figure 2A. D-F) Boxplots showing the diagonal values of the confusion matrices when using only data from a single mark on D) H3K4me1, E) H3K4me3, and F) H3K27ac to annotate global patterns with the model learned using all three marks. Each box corresponds to the global patterns enriched for a chromatin state annotation. The boxes use the same color scheme as the color bars in A-C and Figure 2A.”

“Recovery of global pattern genome annotation using two histone modifications. A-C) Heatmaps displaying the confusion matrices when using only data from pairs of marks: A)

H3K27ac and H3K4me1, B) H3K27ac and H3K4me3, and C) H3K4me1 and H3K4me3 to annotate global patterns based on the model learned using all three marks. Each row corresponds to a state in the original model trained on all three histone modifications. Each column represents the percentage of the genome annotation assigned to each state when using the indicated mark for decoding. The diagonal of the confusion matrix represents the percentage of the original annotation that was recovered based on the pair of marks. The color bar labels each state by the chromatin state annotation similar to Figure 2A. D-F) Boxplots showing the diagonal values of the confusion matrices when using only data from D) H3K27ac and H3K4me1, E) H3K27ac and H3K4me3, and F) H3K4me1 and H3K4me3 to annotate global patterns with the model learned using all pairs of marks. Each box corresponds to the global patterns enriched for a chromatin state annotation. The boxes use the same color scheme as the color bars in A-C and Figure 2A.”

3. The authors compared the chromatin states with gene expression. It will be more interesting to correlate that with coexpression modules, which is another organized expression.

We thank the reviewer for the suggestion. We have added an analysis where the global patterns are correlated with co-expression modules. To perform this analysis, we ran WGCNA on the LCL RNA-seq data resulting in 19 co-expression modules with corresponding eigengene values. We found that the emission parameters of the 85-state model are highly concordant with the module eigengene values, with 16 of the 19 modules significantly associated with one of 72 global patterns represented among these associations. We added a supplementary figure containing three heatmaps, one for each mark, displaying the significance between the emission parameters for the mark in the global pattern and the coexpression module, with significant associations outlined.

The following text was added to the results section:

“We additionally tested for association of the global patterns corresponding to each mark with co-expression modules generated from the subset of individuals for which gene expression was available, which represent patterns in gene expression across individuals (Methods). The association test was performed separately for each mark. This revealed that the module eigengene values are highly concordant with the global patterns with 16 of 19 modules significantly associated with at least one global pattern and 72 global patterns represented among these associations (Linear model, FDR<5%, Figure S11). Thus the global patterns based on epigenetic variation show correspondence with major expression patterns in the data.”

The following text was added to the methods section:

“For the association of global patterns with co-expression modules, we generated co-expression modules from the corrected LCL RNA-seq data using WGCNA [63]. Following WGCNA pre-processing steps, 219 genes were removed due to missing values and one individual was identified as an outlier during clustering and also removed, resulting in 14,450 genes and 53 individuals used for module calculation. We then ran WGCNA which learned a coexpression network consisting of 19 modules. We tested for association between the module eigengene

values and the emission parameters corresponding to each mark. The association was performed separately for each mark. We used an FDR threshold of 5% within each mark.”

“Significance of associations between LCL emission parameters and co-expression modules. The gene expression data was used to identify co-expression modules. The eigengene values of these modules were associated with the global patterns corresponding to each mark. The heatmaps show the $-\log_{10}(\text{FDR})$ of these associations for each global pattern (rows) with each co-expression module (column) for H3K4me1 (left), H3K4me3 (center), and H3K27ac (right). Significant associations (FDR < 5%) are outlined in blue.”

4. It is intriguing to see that the significantly enriched motifs for the TFs are concentrated in only a few states. There could be different TFs responsible for different states. Why most of the states do not have associated TF motifs? Any thoughts?

We thank the reviewer for raising these questions. The significantly enriched TF motifs are concentrated in 20 global patterns due to stringent enrichment requirements. We only report TF motif as enriched in a global pattern if the Fisher Exact two-sided FDR < 5% and the log₂-fold enrichment > 1.5 which restricts the number of global patterns significantly enriched with at least one TF motif. When these requirements are relaxed to a log-fold enrichment > 1, 42 out of the 85 global patterns, corresponding to 98 distinct TFs, are enriched. In our revised version we now show results for significantly enriched TFs for log-fold enrichment > 1 in Supplementary Data 5 and explain that the number of global patterns increases with the relaxed requirements.

Relevant text: “We observed strong enrichment of motifs for 20 of the 85 global patterns, corresponding to 79 distinct TFs (Fisher Exact test, FDR<5%, log₂(fold enrichment) >1.5, Figure 2A), with the number of enrichments increasing to 42 global patterns when we relaxed the enrichment requirements to log₁₀(fold enrichment) >1. These TF motifs were primarily enriched in promoter-like global patterns, with some corresponding to enhancer-like global patterns (Figure 2A, Supplementary Data 5). This motif enrichment in putative regulatory regions of the genome provides orthogonal support of the likely regulatory nature of these genomic regions.”

5. The trans-eQTL replication is done by comparing the p values of the eQTL from LCL and from GTEx. Why not compare effect size, which is less sensitive to sample size?

We thank the reviewer for this suggestion. When considering the effect size and direction of the trans-eQTL replicates, we found that half of the replicates had a significant effect in the opposite direction to the original trans-eQTLs discovered in the LCL dataset. Since we could not replicate the beta value direction using the GTEx dataset, we decided to remove the trans-eQTL analysis. We believe this analysis is best left to future investigations with greater sample sizes.

6. The study did only one replicate assessment for trans-eQTL. Since this study is about the newly-defined chromatin states and related gQTL, it makes more sense to have replicates of these two types of findings.

We have added a replication analysis for the gQTLs using histone modification data from the BLUEPRINT consortium. This data was mapped in H3K27ac and H3K4me1, which overlap with the histone modifications included in the LCL analysis. To perform this replication, we first learned global patterns using this BLUEPRINT data. We trained the stacked ChromHMM models and performed hyperparameter selection in the same way as with the LCL data. The BLUEPRINT data was noisier than the YRI LCL data resulting in a greater number of what we refer to as singleton states where the global pattern corresponds to high histone signal in exactly one individual. To account for the higher level of noise in the BLUEPRINT data, we did not consider these singleton states in the replication analysis. We were able to test the replication of 739 of the 2945 SNPs identified as significant gQTLs in the LCL dataset. We

performed an association test between these relevant SNPs and the BLUEPRINT global patterns and showed that the replicated p-values were more significant than we would expect by chance. We note that this replication is for a variant associated with any global pattern and not of a specific global pattern since the global patterns learned from the two datasets are based on different individuals and thus do not have a direct mapping.

We also note that for the LCL data we showed that global patterns are correlated with large-scale molecular variation such as gene expression and protein quantification data. These associations with different molecular assays not included during model training demonstrates that the individual variation across signals is reproducible and suggests they are capturing biological signal.

The relevant text from the results section is copied below:

“We additionally performed a replication analysis for these identified gQTLs using data from the BLUEPRINT consortium [42]. The p-values obtained in the replication analysis were more significant than expected by chance (Wilcoxon rank sum test $p = 0.03$, Methods, Supplementary Figure S4), supporting that the gQTLs identified using the LCL global patterns contain reproducible signal.”

The relevant text from the figure legend is copied below:

“**LCL gQTL replication analysis.** Results from a replication analysis on 739 of the 2945 gQTLs identified in the LCL data that overlap in the BLUEPRINT dataset. Expected $-\log_{10}(p\text{-values})$ (x-axis) were computed using theoretical values from a uniform distribution. Observed $-\log_{10}(p\text{-values})$ (y-axis) were computed in the replication analysis. The dashed line corresponds to the null, where p-values from the replication experiment have the same quantiles as those from a uniform distribution. The distribution of the replication p-values was significantly lower than we would expect by chance (Wilcoxon rank-sum $p = 0.03$).”

The relevant text from the methods section is copied below:

“We performed a replication analysis on the gQTLs identified in the LCL dataset. We first learned global patterns in BLUEPRINT histone modification data [43]. The histone modification signal was mapped for two marks (H3K27ac and H3K4me1) in 197 individuals from mature neutrophil cells using methods described in [43]. We trained stacked ChromHMM models, identified gQTLs, and selected the number of hidden states using the same methodology as for the LCL data. We used the 80-state model from the BLUEPRINT data for replication as it maximized the number of gQTLs.

We found the BLUEPRINT data to have a greater number of what we refer to as ‘singleton’ global patterns, where the global pattern corresponds to high histone signal in exactly one individual, compared to the YRI LCL data. To increase power in the replication analysis, we restricted the set of SNPs considered for gQTLs in the BLUEPRINT data to those that were associated with [nonsingleton] global patterns. This is analogous to filtering out SNPs with lower minor allele frequency. We identified nonsingleton global patterns as follows. For each global pattern, we counted the number of individuals with emission parameters greater than 0.5 within that global pattern. We defined singleton states as those with only one individual with emission parameter greater than 0.5. All other states were classified as nonsingleton states, including states where all individuals had emissions less than 0.5. Of the 80 global patterns in the final BLUEPRINT model, we classified 54 as nonsingleton states.

Of the 2945 SNPs identified as significant gQTLs in the LCL dataset, 739 were available in the BLUEPRINT dataset. We attempted to replicate each of these SNPs in the BLUEPRINT dataset by associating each with the 54 nonsingleton global patterns. We performed this association test using the same methodology as the LCL gQTL analysis and included the first 20 genotyping PCs as covariates, calculated from the 197 BLUEPRINT individuals using the Plink software version v1.90b6.24 [58]. We compared the p-values obtained in the BLUEPRINT replication with a uniform distribution using a Wilcoxon rank-sum statistical test. We additionally computed the genomic inflation factor to show the increase of significance for replicated SNPs compared to the expected uniform distribution.”

Overall, it is a very exciting new procedure. But some critical tech details like covariate controls will be important to have. More data, comparisons, and assessments are needed to prove the robustness findings, which is the essential proof of the validity and value of such a new approach. It is a very promising work.

We thank the reviewer again for the positive and constructive review. As described above we made a number of revisions to address the reviewers’ concerns as well as comments of other reviewers.

Reviewer #3 (Remarks to the Author):

In this study, Zou and Ernst proposed an framework for identifying and analyzing global patterns in the genome. The global patterns were generated by a stacked chromatin state model by utilizing histone modification data from different samples. The authors applied this framework to two datasets (lymphoblastoid cell lines and autism spectrum disorder, LCL and ASD). They investigated the associations between SNPs, gene expression, protein abundance data, diagnosis status and global patterns. The analysis covers several aspects and is inspiring for other researchers in the epigenomics field. However, I have some concerns about this study.

We thank the reviewer for the summary and the positive comments. We also thank the reviewer for the constructive comments to which we respond to below and led to a substantially improved manuscript.

1) The global patterns framework is constructed based on the stacked chromatin state model and the output global patterns. Vu and Ernst (Genome Biology, 2022) had proposed a "full-stack ChromHMM model", which provides chromatin state annotation by using over 1,000 datasets from more than 100 cell types. Is there any difference between the "stacked chromatin state model" and "global patterns" in this study and "full-stack ChromHMM model" and "universal chromatin states" in Vu and Ernst?

We thank the reviewer for raising this point and we apologize that this was not clearer in our original submission. While both the stacked chromatin state model resulting in learned global patterns in this study and the full-stack ChromHMM model and the learned universal chromatin states in Vu and Ernst are based on training ChromHMM using the stacked approach (Ernst and Kellis 2017), the input data and thus output interpretation is different. For the universal chromatin states, the full-stack ChromHMM model is trained on data from multiple marks across a large number of cell and tissue types, but using data from a very limited number of individuals for each cell or tissue type. The primary focus of this modeling is capturing patterns of epigenetic variation across cell and tissue types. In contrast the stacked chromatin state model presented in this study takes as input data from one or more marks across many individuals in one cell or tissue type. The global patterns learned correspond to combinatorial and spatial patterns across individuals in the epigenetic marks, but are still specific to the singular input cell or tissue type, which in this study was either lymphoblastoid cell line or prefrontal cortex tissue. We distinguish the states modeling human variation in a single cell type as "global patterns" and all other states learned from previous models as "chromatin state annotations" in order to delineate the different training method and state interpretation. We additionally have changed the title to further distinguish our work from previous chromatin state annotations and provide a focus on the recurring patterns of epigenetic variation across individuals that the global patterns represent: "Genome-wide identification and analysis of recurring patterns of epigenetic variation across individuals".

We have modified the introduction to clarify the difference between “global patterns” from this paper and “chromatin states” from previous applications of ChromHMM including the “universal chromatin states” in Vu and Ernst.

“Under this approach, individual-specific genome chromatin state annotations were generated that contain chromatin state assignments for each individual genome-wide.”

“The concatenated approach of the ChromHMM framework, while modeling combinatorial patterns across individuals, does not model any recurring patterns across the genome as each individual is given a unique genome-wide chromatin state annotation.”

“The hidden states of the stacked model are learned on data from multiple individuals in one cell type and represent recurring patterns across individuals and potentially also marks, which we will refer to as ‘global patterns’.”

We have added a sentence in the introduction clarifying the difference between the “stacked chromatin state model” from this study and the “full-stack ChromHMM model” from Vu and Ernst.

“This stacked approach has been previously applied to data from multiple marks across multiple cell types [23] and can be applied in the context of this study across multiple individuals in the same cell type.”

2) The authors analyzed associations between genetic variation, gene expression, protein abundance, diagnosis status and global patterns. Most of these results were numbers of significant QTLs, genes, motifs, and TFs. More detailed results should be provided to convince the audiences.

We thank the reviewer for the suggestion. We have added supplementary data and further explanation addressed in the sub-comments below.

2.1) Tables or lists of these statistically significant QTLs, genes, motifs, and TFs should be provided.

We thank the reviewer for the suggestion. We have added Supplementary Data 5 with information on statistically significant gQTLs, TF motifs, and potential trans-regulators for both the LCL and ASD datasets.

For the gQTLs, we now provide the statistics related to the association test between each global pattern with common variants that are significant based on a Bonferonni corrected threshold. These statistics include the global pattern, SNP, histone modification (for LCL), beta value, t-statistic, p-value, and FDR.

For the TF motif enrichments, we provide the statistics related to the association test between each global pattern and 602 TF motifs. We present the global pattern, TF, fold enrichment, and Fisher-Exact FDR for associations with FDR < 0.05 and a log-fold enrichment > 1. In addition to

the statistics for each significant association between a TF motif and global pattern, we also display the number of TFs significantly enriched in each global pattern along with the names of the TF motifs, the percentage of the genome annotated into each state, and the type of genomic region (i.e. promoter) associated with each global pattern.

For the predicted trans-regulators, we provide the statistics related to the association test between each global pattern with gene expression data for genes where both the TF motifs and expression levels are significantly associated with a pattern. We provide similar statistics as with the gQTL analysis, along with an FDR.

2.2) Some examples of these QTLs and genes, especially functionally related to LCL and ASD, should be showed in the manuscript.

We thank the reviewer for the suggestion. We have created a custom track hub in the UCSC Genome Browser for the LCL data located at <https://github.com/ernstlab/EpiVarIndividuals/>. This resource allows users to view the genome annotation and segmentation along corresponding histone modification data across individuals across the entire genome, allowing investigation of any specific regions of interest. We use this resource to investigate specific trans-regulators, discussed in our response to comment 2.3 below. Relevant text explaining the genome browser tool is provided below.

Relevant text in results: “We visualized the genome segmentation and annotation into global patterns of the final model alongside the histone modification data across individuals in custom genome browser tracks [42].”

Relevant text in data availability: “A UCSC Genome Browser custom track hub for the LCL genome annotation and corresponding histone mark data can be accessed through <https://github.com/ernstlab/EpiVarIndividuals/>.”

We cannot provide a similar browser track for ASD as this data is under restricted access that requires qualified researchers to apply for access to gain approval, however we provide similar visualization of example regions as described in our response to comment 2.4 below.

2.3) The authors identified two TFs that had gene expression patterns associated with the same global pattern for which their motifs were enriched. Examples of expression and histone modification status for specific target genes regulated by these two TFs are needed.

We thank the reviewer for the suggestion. Using the UCSC Genome Browser custom track described in the previous comment, we have visualized example instances of the TP73 and CREB motifs along with histone modification signal data and global pattern annotations. We additionally visualize the association between gene expression and the emission parameters associated with a global pattern for each histone modification using a scatter plot.

Relevant text in results:

“Additionally, the TP73 motif was enriched in an enhancer-like global pattern that is also significantly associated with TP73 gene expression (Figure S14). We show an example instance of the TP73 motif located in a region assigned to this global pattern (Figure S15).”

“We found the CREB5 motif and its gene expression are associated with global patterns associated with active TSS regions (Figure S16, S17).”

Relevant text in supplementary figure legends:

“**Instance of TP73 motif in an enhancer-like global pattern.** Genome browser view showing an instance of TP73 on chromosome 1 located in a segment of the genome annotated to global pattern 14, which is associated with an enhancer chromatin state annotation. The genome browser view shows the TP73 motif present in the region above the global pattern annotations colored based on correspondence to the previous LCL chromatin state annotation in Figure

S5A. Below that is the histone modification signals across individuals for H3K27ac, H3K4me1, and then H3K4me3. H3K4me1 and H3K4me3 can be seen on the following page.”

“**Association between TP73 gene expression and LCL emission parameters.** Scatter plot showing the relationship between ENSG00000078900 (TP73) gene expression (x-axis) and LCL 85-state model global pattern 14 emission parameters (y-axis) for histone modifications H3K4me1 (left), H3K4me3 (center), and H3K27ac (right). Each dot corresponds to a single individual. H3K4me1 and H3K4me3 emission parameters are significantly associated with TP73 gene expression.”

“Instance of CREB5 motif in global pattern associated with Active TSS. Genome browser view showing an instance of CREB5 on chromosome 6 located in a segment of the genome assigned to global pattern 20, which is associated with an active TSS chromatin state annotation. The genome browser view shows genes and a CREB5 motif present in the region, above the global pattern annotations colored based on the previous LCL chromatin state annotations in Figure S5A. Below that is the histone modification signals across individuals for H3K27ac, H3K4me1, and then H3K4me3. H3K4me1 and H3K4me3 can be seen on the following page.”

“Association between CREB5 gene expression and LCL emission parameters. Scatter plot showing the relationship between ENSG00000146592 (CREB5) gene expression (x-axis) and LCL 85-state model global pattern 20 emission parameters (y-axis) for histone modifications H3K4me1 (left), H3K4me3 (center), and H3K27ac (right). Each dot corresponds to a single individual. H3K4me3 emission parameters are significantly associated with CREB5 gene expression.”

2.4) In ASD data, the authors showed that two global states were different in ASD cases and controls. Example regions in the genome should be given.

We thank the reviewer for the suggestion. We have included a new supplementary figure that shows a heatmap of the signal at five 200bp windows that were annotated to the global pattern for the two global patterns whose emission parameters were significantly associated with the ASD diagnosis status (global patterns 47 and 49). Below is the figure legend for the included supplementary figure.

“Example windows annotated into ASD global patterns associated with ASD diagnosis status. The figure shows example 200bp windows of the genome that were annotated into global pattern 47 (top) and 49 (bottom) both significantly associated with the ASD diagnosis status. These example regions shown had the highest Area Under ROC (AUC) values between the corrected histone signal across individuals in a region and the case-control status of the individuals among 200bp windows annotated to the corresponding global pattern. Each heatmap has five rows where each row represents a 200-bp window of the genome annotated with the state of interest and each column represents an individual. The heatmap contents show

the corrected histone modification signal used as input into ChromHMM, with lighter colors representing lower signal. The bar below the heatmap shows diagnosis status of each individual, with individuals sorted by diagnosis status.”

3) The authors integrated multiple sources of public data, including ChIP-seq, RNA-seq, and protein data in LCL and ChIP-seq and RNA-seq data in ASD. This is a good effort, but unfortunately the authors provided three website links for these datasets rather than detailed information.

We thank the reviewers for bringing this to our attention. In the methods, we describe the histone modification data used for the LCL and ASD models, including the exact marks and number of samples. We additionally describe the number of samples available in the RNA-seq and protein data when describing the association tests. Furthermore, we have provided an additional method to access the LCL data for reproducibility purposes. We elaborate on these writing updates, additions to supplementary data, and data hosting in the sub-comments.

3.1) The source link for ChIP-seq and RNA-seq data in LCL is a web server. The data download link on this webserver is currently failing. An updated link and description of the data source should be given.

We thank the reviewers for bringing this to our attention. On multiple occasions, we had been in contact with the authors of the paper providing the original LCL data who would reset the server. However, this was not a permanent solution to the issues with data access. As we do not have access to the web server with the download link, we have hosted the data relevant to our analyses at <https://github.com/ernstlab/EpiVarIndividuals>. We cite the original source and make clear that this hosting location is to ensure reproducibility of our study. We add this link to the data availability section.

“Our work integrates data from a number of published data, including LCL ChIP-seq and RNA-seq data (<http://mitra.stanford.edu/kundaje/portal/chromovar3d/>, a link to a copy of the data we used is available through <https://github.com/ernstlab/EpiVarIndividuals>) [15].”

3.2) The authors analyzed hundreds of high-throughput sequencing datasets. Given that only a subset of the individuals in the LCL data set had corresponding protein data, more information about these data sets should be provided.

We thank the reviewer for this suggestion. We have added Supplementary Data 6 with the list of samples in each of the datasets. We also list the proteins that are significantly associated with at least one global pattern and their corresponding gene expression is significantly associated with at least one global pattern.

3.3) What kind of data file (BAM or fastQ format) have the authors obtained from the public high-throughput sequencing data set? If the authors analyzed fastq format files, the mapping software and parameters should be described in the Methods part.

We thank the reviewer for the question. We used BAMs created by previous publications and thus did not directly analyze FastQ files.

Below is the relevant methods text regarding the data used, which has been updated to include descriptions of the data processing and quality control steps performed on the published data:

“We learned global patterns in LCL histone modification data [15] and ASD histone modification data [4]. In the LCL data, histone modification signal was mapped for three marks (H3K27ac, H3K4me1, H3K4me3) in 75 individuals using procedures described in [15]. Briefly, reads were aligned to personal genomes and filtered for unmapped reads, duplicates, unpaired reads, and reads with low quality. The BAM files from that work were used for this paper. We did not perform any additional quality control beyond their methods. In the ASD data, H3K27ac signal was mapped in the prefrontal cortex tissue for 93 individuals. Reads were mapped to the human reference genome and duplicate reads were filtered. Extensive sample quality control was performed in [15], resulting in a number of samples excluded. We used BAM files for the same 76 individuals that were used in previous work after quality control [4]. We did not perform any additional quality control beyond their methods. Only autosomes were included in this analysis and we used the hg19 genome assembly.”

4) The authors mentioned that "there is greater overlap between these two sets of genes than expected by chance." A simple p-value is not enough. A Venn diagram with specific numbers of each set is needed.

We thank the reviewer for this feedback. We have added a supplementary figure with a Venn diagram of the number of genes significantly associated with gene expression levels and/or protein abundance. Below is the text from the figure caption and the updated text referencing the figure.

Relevant text from results: “After intersection of the protein data with their corresponding gene expression data \cite{Grubert2015}, there were 4319 genes with protein abundance data and gene expression data. Of the 4774 genes with expression patterns significantly associated for at least one global pattern, 1349 also had protein data available. We correlated the abundance of each protein across the 60 individuals with the emission parameters for each global pattern (Methods). We identified 594 proteins that were significantly associated with at least one global pattern (Linear model, FDR < 5%), 588 of which had gene expression data available. Of these 588 proteins, we identified 258 proteins that also had differential expression significantly associated with at least one global pattern (Fisher's Exact test $p=4e-12$, Fold Enrichment 1.4, Supplementary Data 6, Figure S12).”

Intersection of Significantly Expressed Genes
(Fisher Exact p-value 4e-12, Fold Enrichment 1.4)

“Overlap of genes with significant differential expression and significant protein abundance for LCL data. Venn diagram showing the number of genes with gene expression levels, protein abundance, or both significantly associated with at least one global pattern for the LCL data. This analysis only considers the 4319 genes with both protein abundance and gene expression data available with 1349 of these genes having significant differential expression and 588 having significant protein abundance. The overlap of genes between these two sets was significant (Fisher’s Exact Test $p=4e-12$, Fold Enrichment 1.4).”

5) The descriptions of some metrics are lack of clarity.

We thank the reviewer for bringing this to our attention and have revised text to clarify how we compute our metrics.

5.1) The authors used median Spearman correlation to evaluate the robustness of the chromatin state learned from ChromHMM. The description is too simple to confuse me about how it is calculated. A more detailed version of the Methods description is required, especially when a mathematical formula is used.

We thank the reviewer for the feedback. We have updated the text with a more detailed description of this in the methods section with relevant text copied below.

“For each model trained with between 5 to 100 hidden states, we evaluated the pairwise correlations of emission patterns between marks for the same pattern. For each pair of histone modifications (H3K27ac-H3K4me1, H3K27ac-H3K4me3, H3K4me1-H3K4me3), we performed the following analysis to obtain the median Spearman correlation of emission parameters. For each global pattern learned by the model, we computed the Spearman correlation between the emission parameters for the first mark in the pair with the emission parameters for the second mark. After computing this correlation for each global pattern, we took the median Spearman correlation across all global patterns for the model.”

5.2) The descriptions of the “average correlation of histone modification LCL emission parameters and gene expression” in Figure S5-7 are quite confusing. A more detailed description about how to calculate these scores is need.

We thank the reviewer for pointing this out. We have added a paragraph to the Methods section to provide a more detailed description of how we computed the average correlation between global patterns and gene expression as a function of distance.

The revised text in the methods state: “Using the LCL global pattern genome annotation, we determined the correlation of the emission parameters of global patterns with the expression of nearby genes annotated to the pattern. Specifically, for each gene considered from the external RNA-seq data [15], we identified the TSS and divided the genomic region surrounding the TSS into 100bp windows up to 500kb upstream and downstream of the TSS. This resulted in considering 10000 equidistant bins between -500kb and 500kb from the gene's TSS. We then used the LCL genome annotation of global patterns to determine the global pattern assigned to each of these bins. For each bin, we calculated the Pearson correlation between the gene expression and the emission parameters of the bin's assigned global pattern. This was performed separately for the emission parameters of the three histone modifications. We repeated this process for each gene and averaged the Pearson correlations from each bin representing the distance to a gene's TSS and each global pattern.”

References

- Li, X., Lappalainen, T. & Bussemaker, H. J. Identifying genetic regulatory variants that affect transcription factor activity. *Cell Genomics* **3**, 100382 (2023).
- Deplancke, B., Alpern, D. & Gardeux, V. The Genetics of Transcription Factor DNA Binding Variation. *Cell* **166**, 538–554 (2016).
- Claussnitzer, M. et al. Leveraging Cross-Species Transcription Factor Binding Site Patterns: From Diabetes Risk Loci to Disease Mechanisms. *Cell* **156**, 343–358 (2014).
- Flynn, E. D. et al. Transcription factor regulation of eQTL activity across individuals and tissues. *En. PLoS Genet.* **18**, e1009719 (2022).
- Ren, M. et al. Transcription factor p73 regulates Th1 differentiation. *Nature Communications* **11**, 1–12 (2020).
- Wen, A. Y., Sakamoto, K. M. & Miller, L. S. The Role of the Transcription Factor CREB in Immune Function. *The Journal of Immunology* **185**, 6413–6419 (2010).

Reviewers' comments:

Reviewer #1 (Remarks to the Author):

Zou et al has addressed most of the comments. I have remaining minor questions below,

We thank the reviewer again for their previous comments and also thank the reviewer for these additional comments which we respond to below.

1. The authors wrote "For the ASD data set, we accounted for age, sex, percentage of neuronal cells, brain bank, number of peaks, fraction of reads in peak (FRIP), read duplicate fraction, and read alignment fraction [4]." How were "the percentage of neuronal cells" derived and what is the range? Computational deconvolution or from experimental assay?

We thank the reviewer for their questions. According to Sun et al., the percentage of neuronal cells, or neuronal cell fraction, was calculated using CETS [1] from DNA methylation data collected from the same cohort as the ASD histone modification data. This method identifies and uses the top 10,000 cell epigenotype specific (CETS) markers, which are CpGs significantly different between neurons and non-neurons, to quantify the proportions of neurons and glia in observed methylation samples. CETS generates an in silico gradient of DNA methylation for different proportions of glial to neuronal cells and finds the best fit of the observed data to this gradient. The data in Sun et al. has a minimum, mean, and maximum percentage of neuronal cells of 0.092, 0.36, and 0.48, respectively. We have added a citation to the CETS paper when listing the percentage of neuronal cells as a covariate to clarify how this value was calculated.

Relevant text: "For the ASD data set, we accounted for age, sex, percentage of neuronal cells, brain bank, number of peaks, fraction of reads in peak (FRIP), read duplicate fraction, and read alignment fraction [4]. The percentage of neuronal cells was previously determined using the CETS algorithm and DNA methylation data collected in the same cohort [58]."

2. Follow (1), since the cell type composition affects sequencing data significantly, I was wondering what is the trait-trait correlation and whether the traits correlate with the case-control prediction in Figure 4.

We thank the reviewer for this comment. We are assuming that the traits here are referring to the covariates listed above along with the phenotype.

We first considered the association between diagnosis status and the remaining covariates to determine the trait-trait relationships. We found that none of the covariates listed above are significantly associated with diagnosis status (linear model $p < 0.05$).

We next considered the correlations between histone modification signal with covariate values before and after covariate correction to visualize the impact of the correction procedure. The histone modification signal after covariate correction was subsequently used to train the

ChromHMM models. The correlations decreased after covariate correction with the median correlation between corrected histone signal and covariate values ≤ 0.01 as we show in a new supplementary figure.

For the two global patterns previously found to be significantly associated with case-control status shown in Figure 4 (global patterns 47 and 49), we have additionally performed a logistic regression based test while adjusting for covariates to complement the Mann-Whitney test. We applied logistic regression to predict case-control status using the emission parameters for a global pattern along with the covariate values as features resulting in p-values of 0.001 and 0.0004 for the emission parameters of global patterns 47 and 49, respectively. As expected, none of the covariates were significantly associated with case-control status (p-value < 0.05). We previously did not directly account for the covariates in our case-control association test as we already corrected the histone signal for these covariates as described above and were using a Mann-Whitney test. However, we now show that global patterns 47 and 49 are significantly associated with case-control status even when accounting for covariates directly with a logistic regression based test.

Relevant modified text from Results: “In addition to the Mann-Whitney test, we performed logistic regression between global patterns 47 and 49 including covariates and the case-control status (Methods). We found global patterns 47 and 49 to still be associated with ASD status when directly accounting for covariates (p_z-test < 0.05). This was expected as the covariate correction performed on the histone signal used to learn the global patterns removed almost all correlation between signal and covariates (Figure S27).”

Relevant added text to Methods: “We additionally performed logistic regression between the ASD global patterns and the phenotype status. For each global pattern of interest, we performed logistic regression using the emission parameters of the global pattern and the covariates (age, sex, percentage of neuronal cells, brain bank, number of peaks, fraction of reads in peaks (FRIP), read duplicate fraction, and read alignment fraction). These are the same covariates used to correct the histone signal prior to model training. We performed the logistic regression using statsmodels version 0.13.4 Logit function.”

Figure S27. Correlation between histone and covariate signal in ASD dataset before and after covariate correction. Boxplot representing the mean correlation between histone signals and covariates across all individuals before and after covariate correction across histone data. Each dot represents the average correlation across genomic bins within a single chromosome. The covariates Age, Sex, CET (percentage of neuronal cells), Brain Bank, PeakNum (number of peaks), FRIPFract (fraction of reads in peak), DupFract (read duplicate fraction), and AlignFract (read alignment fraction) are accounted for during signal correction. The correlation between signal and covariates is decreased after signal correction.

3. Figure 4 shows the ASD diagnosis with the emission probabilities from the model. Is it possible to have a classifier ROC with this?

We thank the reviewer for the suggestion. We have added a supplementary figure containing the ROC curve for global patterns 47 and 49, which were the two global patterns shown to have a significant association with ASD diagnosis status via the Mann Whitney U test.

Relevant text: “This identified two global patterns (47 and 49) that were significantly associated with ASD status (Mann-Whitney U, $p_{adj} < 0.05$, Methods, Figures 4,S20,S21)”.

Figure S26. ROC Curve for ASD global patterns associated with ASD diagnosis status. ROC curve calculated using the emission parameters for global patterns 47 and 49, both significantly associated with ASD diagnosis status. ROC curve calculated using the emission parameters for each global pattern across all individuals. AUC score is in legend.

Reviewer #2 (Remarks to the Author):

The revision did a great job addressing my previous concerns. Reading through the responses to the other reviewers' comments also made me understand the paper much better. It is a dramatic improvement.

We thank the reviewer again for the review and noting the improvements to the manuscript. We also thank the reviewer for the additional comments to which we respond to below.

Additional questions I have are:

1. In the association test of co-expression modules with emission probabilities, Figure S11, distinct global patterns have been associated with the same module (like ME9) in similar strength. Only a few modules showed a strong association with multiple patterns, while many other modules did not show any association. What does this mean? It is surprising to me. I thought that different global patterns should link to different co-expression modules in a more dramatic way. The figure shows that their (global patterns') co-expression patterns are more similar than different.

We thank the reviewer for these comments and questions. We have added supplementary figures and additional text in the results to further clarify the co-expression analysis. Each of our added analyses and descriptions seek to investigate the properties associated with i) sets of multiple global patterns associated with the same module, ii) global patterns associated with few to no modules, iii) modules that have limited to no association with global patterns, and iv) modules that are strongly associated with many global patterns.

To investigate why modules show strong association with multiple patterns, we added an evaluation of the similarity between global patterns by taking the Pearson correlation between each pair of global patterns and evaluating if the correlations between pairs of global patterns associated with either the same or different co-expression modules differ. If a pair of global patterns is associated with the same module, then the emission parameters are likely to be more highly correlated than a pair of global patterns not associated with the same module. We confirmed this via a boxplot and Wilcoxon rank-sum test demonstrating that a pair of global patterns that are significantly associated with the same module tend to be more highly correlated than pairs of global patterns that are not.

We additionally added a colorbar to S11 to visualize the chromatin state with the highest fold enrichment for each global pattern to evaluate if the type of genomic region represented by a global pattern relates to its associations with co-expression modules. This demonstrates that groups of global patterns associated with few to no co-expression modules (such as patterns 45-49 and 71-81) tend to have lower signal and are associated with repressed polycomb, heterochromatin, and transcribed chromatin states. While global patterns associated with promoter and enhancer chromatin states, which represent most global patterns, tend to be associated with multiple co-expression modules.

To investigate the properties of sets of modules that are associated with multiple global patterns compared to sets of modules that are associated with few to no global patterns, we added an analysis investigating the behavior of genes that are assigned to each co-expression module. Specifically, we analyzed whether the expression of genes assigned to a specific module are more or less likely to be associated with global patterns when tested directly. For each co-expression module, we plotted the number of global patterns that module is significantly associated with against the percentage of genes assigned to the module that are significantly associated with at least one global pattern. As a global pattern has emission parameters for each of the three histone marks (H3K4me1, H3K4me3, H3K27ac), we performed this analysis on a per-mark basis. This scatter plot shows that co-expression modules containing a higher percentage of genes that are directly associated with at least one global pattern are more likely to be associated with multiple global patterns. Similarly, if a limited number of genes are associated with global patterns in a module, that module will likely be associated with few to no global patterns.

We additionally investigated the relationship between the number of genes significantly associated with a specific global pattern within a specific co-expression module and the strength of the association between that co-expression module and global pattern. We again divided this analysis on a per-mark basis corresponding to the global pattern emission parameters for each mark. For this, we plotted the number of genes significantly associated with a global pattern against the $-\log_{10}(\text{FDR})$ of the association between a co-expression module and global pattern. For each module plot, one point represents a single global pattern. Generally, the greater the number of genes significantly associated with a global pattern within a co-expression module, the greater the strength of the association between the co-expression module and global pattern.

The relationship between global patterns and co-expression modules being associated with the behavior of the specific genes within each module is expected as a co-expression module captures shared expression patterns across genes. This relationship along with the existence of approximately one-third of genes considered in the study significantly associated with at least one global pattern is consistent with that certain modules are strongly associated with many global patterns. Conversely, the existence of a substantial number of genes not associated with any global patterns is consistent with other modules not being associated with any global patterns. Generally, both the gene expression and co-expression module association tests demonstrate that global patterns are associated with large-scale expression patterns, despite expression data not being used during model training.

The beginning of the first paragraph below is the co-expression analysis from the previous submission (written in a darker orange to denote highlight when the new text begins). We have now substantially expanded on this text adding explanations for the reviewer's questions. We have also added the module eigengenes and list of genes assigned to each module to Supplementary Data 5.

Relevant text from co-expression results: “We additionally tested for association of the global patterns corresponding to each mark with co-expression modules generated from the subset of individuals for which gene expression was available, which represent patterns in gene expression across individuals (Methods). The association test was performed separately for each mark. This revealed that the module eigengene values are highly concordant with the global patterns with 16 of 19 modules significantly associated with at least one global pattern and 72 global patterns represented among these associations (Linear model, FDR<5%, Figure S11, Supplementary Data 5). Among global patterns that have a significant association with at least one co-expression module, 62.5% were most associated with enhancer and promoter chromatin states. The remaining global patterns not associated with a module tend to have lower signal and were most associated with repressed polycomb, heterochromatin, and transcribed states (Figures 2A,S11). Additionally, certain groups of global patterns tended to be associated with the same co-expression module consistent with inter-pattern correlation. Specifically, global patterns that are associated with the same co-expression module tend to have substantially higher pairwise correlation between emission parameters than patterns not significantly associated with the same module (Figure S12, Wilcoxon rank-sum p-value < 0.001).

Certain co-expression modules exhibited strong association with multiple patterns while other modules exhibited few to no associations with any global pattern. The difference in behavior between modules is associated with the genes assigned to each module. As stated previously, approximately one-third of the genes considered in this study are significantly associated with at least one global pattern, with all 85 global patterns included among these associations and certain genes being associated with multiple global patterns. Co-expression modules with a higher percentage of genes included in this subset of significant genes are more likely to be associated with multiple global patterns (Figure S13). Moreover, the strength of the association between co-expression modules and multiple global patterns is associated with the number of genes associated with a global pattern within a co-expression module (Figure S14-S17). This relationship between global patterns and co-expression modules being associated with the behavior of specific genes within each module is expected as a co-expression module captures shared expression patterns across genes. The relationship between co-expression modules and the genes they represent along with the existence of a substantial number of genes significantly associated with at least one global pattern is consistent with that certain modules are strongly associated with global patterns. Conversely, the existence of a substantial number of genes not associated with any global patterns is consistent with other modules not being associated with any global patterns. Overall, these results from both the gene expression and co-expression module association tests demonstrate that global patterns of epigenetic variation are associated with large-scale expression patterns, despite expression data not being used during model training.”

Figure S11. Significance of associations between LCL emission parameters and co-expression modules. The gene expression data was used to identify co-expression modules. The eigengene values of these modules were associated with the global patterns corresponding to each mark. The heatmaps show the $-\log_{10}(\text{FDR})$ of these associations for each global pattern (rows) with each co-expression module (column) for H3K27ac (left), H3K4me1 (center), and H3K4me3 (right). Significant associations (FDR < 5%) are outlined in blue. Each global pattern is annotated (left) with a previous LCL chromatin state annotation for one individual [49] with the highest significant enrichment (color bar legend on right).

Figure S12. Similarity between LCL global patterns significantly associated with the same or different co-expression modules. Boxplots representing the pairwise correlations between emission parameters for all LCL global patterns. Emission parameters for the histone marks (H3K4me1, H3K4me3, and H3K27ac) are visualized separately. The pairwise correlations are further divided whether the pair of global patterns is significantly associated with the same or different co-expression module. 'Same' represents two global patterns that are significantly associated with the same co-expression module. 'Different' represents two global patterns that are significantly associated with different co-expression modules or are not significantly associated with any co-expression module.

Figure S13. Relationship between the percentage of significant genes within a co-expression module and the number of global patterns associated with the co-expression module. The scatterplots show the relationship between co-expression modules and the genes assigned to each module. Plots provided per mark for H3K4me1, H3K4me3, and H3K27ac (left to right). Each point represents a co-expression module. X-axis is the number of

global patterns with emission parameters significantly associated with a co-expression module's eigengenes. Y-axis is the percentage of genes assigned to a co-expression module with expression significantly associated with at least one global pattern.

Figure S14. Relationship between the number of significant genes within a co-expression module and the strength of association between a co-expression module and global pattern. Each scatterplot shows the relationship between the number of genes significantly associated with a global pattern in a co-expression module and the strength of the association between the co-expression module and global pattern. Each point represents a global pattern. X-axis is the number of genes with expression significantly associated with a global pattern's emission parameters. Y-axis is the $-\log_{10}(\text{FDR})$ of the association between a co-expression module's eigengenes and a global pattern's emission parameters. Associations based on each mark (H3K4me1, H3K4me3, and H3K27ac) are colored separately within each plot. Horizontal line represents significance threshold ($\text{FDR} < 5\%$) for association test between co-expression module eigengenes and global pattern emission parameters. The title of each subplot corresponding to a single co-expression module contains the Pearson correlation between the number of significant genes and the $-\log_{10}(\text{FDR})$ of the module-global pattern association for H3K4me1 (left), H3K4me3 (middle), and H3K27ac (right).

Number of genes significantly associated with global patterns and assigned to coexpression modules: H3K4me1

Figure S15. Number of genes significantly associated with global patterns within co-expression modules for H3K4me1. For each combination of co-expression module and global patterns, the number of genes assigned to the co-expression module with expression significantly associated with the global pattern's emission parameters for H3K4me1. Each column represents genes assigned to a co-expression module. Each row represents a global pattern. A single gene may be associated with multiple global patterns. The number in each box represents the number of genes significantly associated with a global pattern, matching the color scale from light to dark blue. Boxes are outlined if the co-expression module is significantly associated with a global pattern (same as Figure S11).

Number of genes significantly associated with global patterns and assigned to coexpression modules: H3K4me3

Figure S16. Number of genes significantly associated with global patterns within co-expression modules for H3K4me3. For each combination of co-expression module and global pattern, the number of genes assigned to the co-expression module with expression significantly associated with the global pattern's emission parameters for H3K4me3. Each column represents genes assigned to a co-expression module. Each row represents a global pattern. A single gene may be associated with multiple global patterns. The number in each box represents the number of genes significantly associated with a global pattern, matching the color scale from light to dark blue. Boxes are outlined if the co-expression module is significantly associated with a global pattern (same as Figure S11).

Number of genes significantly associated with global patterns and assigned to coexpression modules: H3K27ac

Figure S17. Number of genes significantly associated with global patterns within co-expression modules for H3K27ac. For each combination of co-expression module and global pattern, the number of genes assigned to the co-expression module with expression significantly associated with the global pattern's emission parameters for H3K27ac. Each column represents genes assigned to a co-expression module. Each row represents a global pattern. A single gene may be associated with multiple global patterns. The number in each box represents the number of genes significantly associated with a global pattern, matching the color scale from light to dark blue. Boxes are outlined if the co-expression module is significantly associated with a global pattern (same as Figure S11).

2. I am also surprised to see that a few individual genes, like TP73, correlated with global pattern 14. A global pattern should be more related to global regulation instead of local, regional regulation, which is related to one target gene. Certainly the correlation is not that strong. Therefore, its meaning may not be that important. The co-expression module might be more relevant and, therefore, deserves more space in the main text.

We thank the reviewer for raising this point and apologize that the purpose of our focus on a few individual genes was not clear. Each global pattern, including global pattern 14, is associated with the expression of many different genes. Specifically, we found 4,744 genes (approximately one-third of the genes investigated in this study) were significantly associated with at least one global pattern, demonstrating that global patterns co-vary with a large number of genes and can represent patterns of either global or local regulation. We discuss a few individual genes as examples of potential trans-regulators related to our TF motif enrichment analysis. In this analysis, we computed the motif enrichment of 602 TF motifs and found 79 distinct motifs are enriched in at least one global pattern. We found that 24 of these 79 TF motifs also had their expression levels significantly associated with a global pattern, with two TFs (TP73 and CREB5) having their gene expression associated with the same global pattern for which their motifs were enriched. This analysis was meant as an example of how global patterns could be used to identify potential trans-regulators and not to reflect how the global patterns are related to patterns of global regulation. Our other analyses comparing the global patterns to data not included during training such as gene expression, co-expression modules, and protein quantification demonstrate the relationship between global patterns and large-scale molecular variation and regulation.

We have updated the text in the TF motif and trans-regulator prediction analysis to clarify that the focus on the individual genes TP73 and CREB5 does not reflect the global patterns relation to global regulation and the expression of a large number of genes: “We highlight two TFs (TP73 and CREB5) which had gene expression patterns associated with the same global patterns for which their motifs were enriched. These genes provide examples of how global patterns can be used to identify potential trans-regulators supported by multiple types of evidence, but we note many other genes have their expression levels significantly associated with the global patterns.”

Additionally, we have added the results of the association test between gene expression and global pattern emission parameters in Supplementary Data 5 to further highlight the quantity of genes associated with patterns of epigenetic variation, where we previously provided only the gQTL analysis and TF motif enrichment results. We have also added supplementary figures visually demonstrating the breadth of the association between gene expression and global patterns. These figures break down the number of genes that are significantly associated with each global pattern, divided by histone mark and the co-expression module to which the genes are assigned. These supplementary figures are included in our co-expression analysis described in our response to comment 1 above.

In addition to the supplementary figure described above, we have also added additional analyses related to the relationship between global patterns and co-expression modules and expanded the discussion of co-expression modules in the main text. These updates are described in detail in our response to comment 1 above. We additionally now provide the module eigengenes and the list of genes assigned to each module in Supplementary Data 5.

3. Do gQTLs match any eQTLs or other QTLs in LCLs? GREAT analysis did not answer this question.

We thank the reviewer for this question. We have added an analysis investigating the overlap between gQTLs and eQTLs in the LCL dataset. We performed an association test between SNPs included in the set of gQTLs and all genes available in RNA-seq datasets for overlapping individuals. These are the same datasets used for the association tests between global patterns and gene expression. For this analysis, we first consider the full set of 2,945 gQTLs, which in our methods have an MAF > 5% when considering all individuals in the genotyped dataset. However, since gene expression data is only available in a subset of 54 individuals, we additionally investigated a subset of 1,016 gQTLs with an MAF > 5% when only considering the overlapping individuals available in the gene expression data. For both sets of gQTLs considered, we additionally performed the eQTL analysis on 100 random permutations of SNPs with the same allele frequency distribution as the gQTL set. When considering all SNP-gene pairs identified in the eQTL analysis, the number of significant eQTLs found in the gQTL set was greater than the number of eQTLs found in any of the random SNP permutations for both the complete set of gQTLs and subset of common gQTLs in the set of gene expression individuals. However, in the eQTL set certain genes are associated with multiple unique SNPs which could inflate the significance of the SNP overlap. We investigated the gQTL-eQTL overlap when only considering the top most associated SNP for each unique gene in the eQTL set. On the full set of gQTLs, the number of significant eQTLs from this filtered set found in the gQTL set (38 unique SNPs) was greater than 96% of the number of eQTLs found in any of the random SNP permutations. For the subset of gQTLs based on the gene expression individuals (28 unique SNPs), 90% of the number of eQTLs found in any of the random SNP permutations. These results suggest that gQTLs may contain biologically relevant signals related to gene expression and can potentially be used to study regulatory effects.

Relevant text from LCL results: “We further explored the gQTLs identified in the LCL cell type by conducting an eQTL analysis between the genotype information and external gene expression [15] in the subset of individuals (n=54, Supplementary Data 6) for which gene expression data was available. We associated both the full set of gQTLs and the subset of gQTLs with MAF > 5% considering this subset of individuals with gene expression data available with all available genes. We did the same for random permutations of SNPs and identified the set of significant eQTLs (Bonferonni corrected p-value < 0.05) when considering only the top-most associated SNP for each unique gene for the eQTL set. We observed that the number of significant eQTLs from this filtered set found in the gQTL set was greater than 96% of the number of eQTLs found in any of the random SNP permutations when considering the full set of gQTLs. The number of overlapping significant eQTLs was greater than in 90% of permutations when considering the

subset of common variant gQTLs based on the subset of individuals with gene expression data available at MAF > 5% (Methods). These results suggest that gQTLs may contain signal related to gene expression and help identify cis- and trans-regulatory effects.”

Relevant text from Methods: “For the LCL dataset, we used the available genotypes and external RNA-seq data to perform an eQTL analysis and determine the overlap between gQTLs and the identified eQTLs. Expression data was previously collected for 54 of the 75 samples using RNA-seq [15] (Supplementary Data 6). We corrected the expression data for known covariates (sex, genotyping method, number of reads, the ratio between the fragment-length peak and the read-length peak or relative strand correlation (RSC) [61]). We considered the same set of genes as in a previous publication [15].

To compute eQTLs, we computed the allele frequency of each gQTL both considering all available individuals and only considering the subset of 54 individuals with gene expression data available and identified a subset of 1,016 gQTLs with MAF > 5% based on this subset. For both the full gQTL set and subset of gQTLs, we generated 100 random permutations of the same size and allele frequency distribution as the gQTL set. We performed an association test between the gQTL sets plus the corresponding 100 random permutations for each set and the gene expression data to identify eQTLs. We calculated the association statistics for a linear model using the MatrixEqtI v2.2 software [59] and used a Bonferonni corrected p-value of 0.05 to identify significant eQTLs. We considered the top-most significant SNP for each unique gene in the eQTL set. The number of gQTLs that resulted in a significant eQTL by this threshold was compared to the number of significant eQTLs identified in each random SNP permutation.”

4. The paper used 200bp non-overlapping bin to run ChromHMM. Is there any concern regarding the effects of the random breakdown of a continuous sequence? Will the results change with the boundaries of bins? Will sliding window give a different result? Maybe I still have missed something. Hope the authors can help to clarify.

We thank the reviewer for the questions. The approach of using 200bp non-overlapping intervals is the default of ChromHMM and has previously been applied effectively in many other applications. As 200bp corresponds to roughly the size of a nucleosome and flanking region and neighboring nucleosomes often have similar histone modifications, working at this resolution with a binning approach has relatively minimal loss of information. There are also a number of advantages built into this approach including: (1) more efficient model learning and inference compared to working at a fine resolution; (2) input consistent with HMM assumptions that the observation at a position only depends on the hidden state at the position; (3) a model less susceptible to overfitting local signal variation such as dips of histone signal because of nucleosome depleted regions; (4) ensures that no segment is excessively short. While other related tools have made different design decisions that might be closer to what the reviewer is suggesting by a sliding window, we believe that these advantages contribute to the effectiveness and popularity of ChromHMM in this context.

We have added text to the Methods to clarify the choice of a 200bp bin: “The approach of using non-overlapping bins of 200bp length, corresponding to roughly the size of a nucleosome and flanking region, follows the default for ChromHMM and what has been effectively applied in many prior applications of ChromHMM.”

Reviewer #3 (Remarks to the Author):

My comments have been addressed.

We thank the reviewer again for their review.

References

[1] Guintivano, J., Aryee, M. J. & Kaminsky, Z. A. A cell epigenotype specific model for the correction of brain cellular heterogeneity bias and its application to age, brain region and major depression. *Epigenetics* 8, 290–302 (2013).